

# Assimilating aerosol optical properties related to size and absorption from POLDER/PARASOL with an ensemble data assimilation system

Athanasios Tsikerdekis [1,2], Nick A.J Schutgens [2], Otto P. Hasekamp [1]

[1]SRON Netherlands Institute for Space Research, Utrecht, the Netherlands
     [2]Department of Earth Science, Vrije Universiteit Amsterdam, 1081 HV Amsterdam, the Netherlands

*Correspondence to*: Athanasios Tsikerdekis (A.Tsikerdekis@sron.nl)

**Abstract.** A data assimilation system for aerosol, based on an ensemble Kalman filter, has been developed for the global aerosol model ECHAM-HAM and applied to POLDER derived observations of optical properties. The advantages of this
assimilation system is that the ECHAM-HAM aerosol modal scheme carries both aerosol particle numbers and mass which are both used in the data assimilation system as state vector, while POLDER retrievals in addition to Aerosol Optical Depth (AOD) and Angstrom Exponent (AE) provide also information related to aerosol absorption like Aerosol Absorption Optical Depth (AAOD) and Single Scattering Albedo (SSA). The developed scheme can assimilate simultaneously combinations of multiple variables (e.g. AOD, AE, SSA), to optimally estimate mass mixing ratio and number mixing ratio of different
aerosol species. We investigate the added value of assimilating AE, AAOD and SSA, in addition to the commonly used AOD, by conducting multiple experiments where different combinations of retrieved properties are assimilated. Results are evaluated with (independent) POLDER, MODIS Dark Target, MODIS Deep Blue and AERONET observations. The experiment where POLDER AOD, AE and SSA are assimilated shows systematic improvement in mean error, mean absolute error and correlation for AOD, AE, AAOD and SSA compared to the experiment where only AOD is assimilated.
The same experiment reduces the global ME against AERONET from 0.072 to 0.001 for AOD, from 0.273 to 0.009 for AE and from -0.012 to 0.002 for AAOD. Additionally, sensitivity experiments reveal the benefits of assimilating AE over AOD at a second wavelength or SSA over AAOD, possibly due to a simpler observation covariance matrix in the present data assimilation framework. We conclude that the currently available AE and SSA do positively impact data assimilation.

## 1. Introduction

Atmospheric aerosol is a key factor that modifies the effects and intensity of climate change, due to its participation in numerous atmospheric processes that may alter the radiative budget of the planet (Boucher et al., 2013). The aerosol diverse size and chemical composition, that affects their transport and removal mechanisms, the complex atmospheric aging processes, the in-cloud condensation growth and the limited as well as indirect information regarding their emission flux and sources, makes their simulation a challenging task (Huneeus et al., 2011; Kinne et al., 2006; Schutgens and Stier, 2014;



Textor et al., 2006) and the aerosol direct, semi-direct and indirect radiative effect very hard to estimate (Carslaw et al., 2013; Fan et al., 2016; Hasekamp et al., 2019b; Koch and Del Genio, 2010; Myhre et al., 2013; Nabat et al., 2014; Tsikerdekis et al., 2017, 2019; Yumimoto and Takemura, 2011).

Data assimilation systems have been employed in the past in order to either adjust the aerosol mixing ratio (Benedetti et al., 2009; Dai et al., 2014, 2019; Escribano et al., 2017; Schutgens et al., 2010a, 2010b; Di Tomaso et al., 2017; Yumimoto et al.,

2007, 2016) or estimate new aerosol emission fluxes (Chen et al., 2018, 2019; Huneeus et al., 2012; Pope et al., 2016; Schutgens et al., 2012; Sekiyama et al., 2010; Xu et al., 2013).

Retrieved Aerosol Optical Depth (AOD) from the Moderate Resolution Imaging Spectroradiometer dark target algorithm (MODIS-DT) has been extensively assimilated (Benedetti et al., 2009; Dai et al., 2014; Escribano et al., 2017; Huneeus et al., 2012; Schutgens et al., 2012; Di Tomaso et al., 2017; Xu et al., 2013; Yumimoto and Takemura, 2011), or used for

validation as independent observations (Dai et al., 2019; Schutgens et al., 2010a, 2010b) in past studies. Studies focused on dust AOD or dust source regions assimilated AOD retrievals from the MODIS dark blue algorithm (MODIS-DB) (Escribano et al., 2017; Di Tomaso et al., 2017), while other studies assimilated AOD from the ground-based network of stations AERONET (Schutgens et al., 2012, 2010a, 2010b) or AOD from the Himawari-8 (Yumimoto et al., 2016, 2018). Retrieved AOD from POLDER GRASP algorithm (Dubovik et al., 2011) was also assimilated by Chen et al. (2019) and Escribano et

al. (2017).

Almost all aerosol assimilation systems assimilate Aerosol Optical Depth (AOD). AOD is a quantity that describes the aerosol extinction (scattering + absorption) in the total column of the atmosphere, and is, if all microphysical properties stay the same, related to the amount of aerosols. AOD is affected by the size and absorption of aerosol particles, but assimilating just AOD in one wavelength does not disentangles fine from coarse particles and absorbing from non-absorbing particles.

The assimilation of other satellite retrieval products, along with AOD, like Angstrom Exponent (AE) and Single Scattering Albedo (SSA), which are more closely linked to size and absorption of aerosol particles, may have a positive impact on data assimilation.

The importance of assimilating total and fine mode AOD separately were highlighted by Generoso et al. (2007). MODIS total and fine mode fraction AOD were assimilated by Dubovik et al. (2008) and Huneeus et al. (2012), while recently total

and fine mode fraction AOD from MODIS were assimilated simultaneously for dust only simulations by Escribano et al. (2017). Assimilating species-specific observations, like dust AOD from LIVAS (Amiridis et al., 2013), may also address dust related biases in the MACC reanalysis (Georgoulias et al., 2018). Benefits on aerosol size correction was demonstrated also by assimilating simultaneously AODs in two wavelengths or preferably AOD & AE (Schutgens et al., 2010a, 2010b). In addition, even for remote sensing measurements with relatively high uncertainty on the light absorbing properties, particle

size related information and the particle absorbing properties proved to be highly beneficial for a better representation of aerosol composition (Chen et al., 2019). However, while the AOD is retrieved for at least one wavelength from all satellites, the other observational parameters can be retrieved only by a limited number of remote sensing instruments (e.g. POLDER (Dubovik et al., 2011; Hasekamp et al., 2011), OMI (Torres et al., 2007).





Remote sensing is the best way of obtaining aerosol observations over large regions of the Earth. Satellites instruments do not directly measure aerosol related information, but properties of light such as intensity, colour, polarization state (Benedetti et al., 2018). Although the direct assimilation of clear-sky radiance has been attempted in the past (Weaver et al., 2007), a typical aerosol data assimilation system assimilates aerosol optical properties, which are obtained using retrieval algorithms that use as input the satellite clear-sky measurements. Satellite sensors and retrieval algorithms introduce uncertainties in these estimates, due to satellite radiometric calibrations, aerosol properties assumptions, cloud contamination and surface albedo diverse characteristics (Li et al., 2009). In a data assimilation system, the uncertainty of observations should be defined and given as input.

Global aerosol simulations have shown that the aerosol atmospheric load and microphysical properties between different climate models, or even within the same model with altered parameterizations, is quite diverse (Huneeus et al., 2011; Miller et al., 2006). An ensemble based data assimilation system uses an ensemble of perturbed simulations to define the uncertainty in states of aerosol in the atmosphere (Schutgens et al., 2010a). This ensemble of simulations can be then adjusted based on aerosol retrievals, to derive a new better estimate of aerosol state, which is represented by the ensemble mean.

A widely used data assimilation method that combines an ensemble of perturbed simulations (a-priori or background) and provides a new better estimate (a-posteriori or analysis) is the Local Ensemble Transform Kalman Filter (LETKF) (Hunt et al., 2007; Miyoshi and Yamane, 2007). In order to get a new better estimate of aerosol state, which in our case is defined as the atmospheric aerosol mass mixing ratio and number mixing ratio, LETKF requires two main ingredients. (1) An estimate of the background state and the associated uncertainty; (2) observations and the associated uncertainty. The observations must be related to the state vector (e.g. aerosol mixing ratio), the state vector is converted to simulated observations by a model (in our case ECHAM-HAM), while the background estimate and its uncertainty may be represented through an ensemble of simulations.

In the present study, we use LETKF to assimilate aerosol optical properties (AOD$_{550}$, AOD$_{865}$, AE$_{550-865}$, AAOD$_{550}$, SSA$_{550}$; subscript denotes wavelength in nm) from multiangle photo-polarimetric POLDER measurements retrieved by the algorithm developed at SRON Netherlands Institute for Space Research. In Section 2, we present POLDER retrievals and the corresponding uncertainty estimate, the global climate-aerosol model ECHAM-HAM, as well as other observational data used as independent observations. Section 3 presents the data assimilation system, the method to produce the perturbed ensemble necessary for LETKF and the experimental set up. Finally, in Section 4, the results are partitioned into four distinct segments. The first highlights the importance of combining aerosol optical properties (e.g. AOD & AE & SSA) to acquire a more robust representation of the atmospheric aerosol state, the second evaluates the core-experiments with independent observations, the third discusses the preference of some observations over others for the assimilation, and lastly a number of sensitivity-experiments is presented regarding some of the parameters in LETKF.



## 2. Data

### 2.1 Observational Data

#### 2.1.1 AERONET

AERONET is a global network of ground based stations of Sun-sky radiometers (Holben et al., 1998) that provides high
quality Direct-Sun AOD estimates at various wavelengths from 340 to 1020nm (Holben et al., 2001) and Aerosol Inversion
SSA estimates (Dubovik and King, 2000). Due to its design the instrument can provide useful measurements under cloud
free conditions during the day. The AOD uncertainty is estimated to be <0.02 (Dubovik et al., 2000; Eck et al., 1999) and
SSA < 0.03 for $AOD_{440}>0.4$ and Solar Zenith Angle greater than 50° (Dubovik et al., 2002; Holben et al., 2006). Thus,
AERONET is commonly used as the "ground truth" for the validation of aerosol optical properties, in particular AOD, in
both satellite and model studies alike.

For the evaluation of the assimilated experiments the V3 L2.0 datasets of the Direct-Sun and Aerosol Inversion data sets are
used. Recently AOD monthly differences between AERONET V3 and V2 were less than 0.002±0.002, highlighting the
stability of the network independent of the version (Giles et al., 2019). It is noted that for aerosol absorption properties.
AERONET L2.0 Aerosol Inversion data only includes cases with $AOD_{440} > 0.4$, thus the evaluation of ECHAM-HAM
absorbing properties is restricted to high AOD cases only. To define the POLDER uncertainty V3 L1.5 Inversion data sets
were used because it provides more data points at the cost of accuracy. AERONET L1.5 contains cases of low $AOD_{550}$
where the retrieval accuracy for AAOD and SSA is low.

#### 2.1.2 POLDER

The POLDER-3 instrument on the Polarization and Anisotropy of Reflectances for Atmospheric Sciences coupled with
Observations from a Lidar (PARASOL) micro-satellite, which has been active between 2004 and 2013, has the unique
capability of measuring light intensity and polarization properties at multiple viewing angles (up to 16) and multiple
wavelengths (0.44 to 1.02μm). The multi-wavelength and multi-viewing-angle photopolarimetric measurements make better
use of the information content of scattered solar radiation in comparison to single-viewing measurements (Hasekamp and
Landgraf, 2007; Mishchenko and Travis, 1997), hence POLDER is an ideal tool for obtaining accurate aerosol microphysical
and optical properties. Its native horizontal resolution is $6 \times 6$ km$^2$ but in this study 'Medium Resolution' data have been used
that correspond to $18 \times 18$ km$^2$. The retrieval algorithm developed at SRON - Netherlands Institute for Space Research, fits a
radiative transfer model (Hasekamp, 2005; Schepers et al., 2014) to the multiangle photopolarimetric measurements of
POLDER to derive aerosol optical properties corresponding to a bi-modal aerosol size distribution. The retrieved properties
for two modes for fine and coarse particles are the effective radius and effective variance, the column number concentration
and the real and imaginary part of the refractive index for each mode, and for the coarse mode additionally the fraction of
spherical particles is retrieved (Hasekamp et al., 2011; Lacagnina et al., 2015; Wu et al., 2015). Using the abovementioned





aerosol parameters, for the two modes, Aerosol Optical Depth (AOD), Angstrom Exponent (AE), Absorption optical Depth (AAOD) and Single Scattering Albedo (SSA) can be calculated. The multi-angle multi-wavelength photopolarimetric measurements of POLDER also have the ability to differentiate scattering of cloud droplets from aerosol particles, making

possible to exclude cloud contaminated pixels (Stap et al., 2015). Recently the algorithm was extended to an arbitrary number of modes (Fu and Hasekamp, 2018) but the present paper uses the bi-modal product. In the present study aggregated (1° × 1°) POLDER data are used in the assimilation. Global aerosol retrievals from POLDER-3 by the SRON algorithm are available for the year 2006.

The aerosol optical properties of POLDER retrievals demonstrate good agreement with either ground based (AERONET) or

satellite (Ozone Monitoring Instrument; OMI) retrievals for the year 2006 (Hasekamp et al., 2011; Lacagnina et al., 2015, 2017; Stap et al., 2015). A global evaluation against AERONET inversion dataset for POLDER ocean and land pixels showed similar results, with absolute differences for three AOD wavelengths and SSA between ±0.05 (Lacagnina et al., 2015, 2017). Performance over land and ocean pixels is similar for SSA, contrary to AOD where retrievals over ocean pixels were better (Lacagnina et al., 2017). POLDER AOD agreement with AERONET descents considerably for values below

0.07 over land, which results in some bias spatial patterns, where in low AOD regions like North America (relative) errors were large and in high AOD regions like Sahara errors were small. SSA largest relative discrepancies are found in Europe and North America (Lacagnina et al., 2017). The recent multi-model version of the algorithm (10 modes instead of 2 modes) achieved higher accuracy for AOD and similar performance for SSA when compared to AERONET for retrievals over land (Fu and Hasekamp, 2018).

In the present study, the observational uncertainties of POLDER are assessed by evaluating the retrieval against the dataset of AERONET. This approach provides a parameterization of observational uncertainty, based on the real errors of POLDER retrievals. Obviously, since AERONET is a spatially sparse ground-based network of stations some generalization had to be made as far as the performance of POLDER in remote areas. Furthermore, the AERONET retrieval errors, which typically are smaller than the errors of any remote sensing retrieval, at least for AOD, were not taken into account. The detailed

description for the POLDER uncertainty estimation is presented in Appendix A.

Global maps of the resulting absolute and relative uncertainties of POLDER averaged over 40 days (the period of our assimilation experiment) for the five variables used in the present study are illustrated in Figure 1. Also, the AERONET stations used are selected for POLDER retrievals over land, where the errors are higher in comparison to POLDER retrievals over ocean (Lacagnina et al., 2017). For the above-mentioned reasons, in cases of very low $AOD_{550}$ (for example in the first

two $AOD_{550}$ bins of FigureA 1 and predominantly over ocean pixels, the POLDER uncertainty estimates are probably too conservative. The POLDER uncertainty estimation is based on spatiotemporal POLDER retrievals of a $18 \times 18$ $km^2$ grid with AERONET. The estimated uncertainty was afterwards applied to a coarser 1° × 1° grid.

The global mean absolute uncertainties for AOD in the two wavelengths (550nm, 865nm) is quite similar (0.08, 0.06), although $AOD_{865}$ is lower since the absolute values of $AOD_{865}$ are lower too. The $AE_{550-865}$ global mean absolute uncertainty

is 0.50, for $SSA_{550}$ 0.085, and for $AAOD_{550}$ 0.012. It can be clearly seen that the error on $AE_{550-865}$ and $SSA_{550}$ strongly





depend on AOD. For example, the error on SSA is ~0.03 is regions with high aerosol loading due to biomass burning, dust outbreaks, or industrial pollution but can be ~0.10 over the remote ocean.

### 2.1.3 MODIS-DT and MODIS-DB

For a broader spatial coverage, a comparison with the independent observations MODIS Collection 5 Dark Target (MODIS-
DT) and MODIS Collection 6 Deep Blue (MODIS-DB) retrievals of $AOD_{550}$ (Sayer et al., 2014) was conducted. This allows assessment of the performance of the assimilated experiments over ocean and other remote regions, away from AERONET sites. In the case of the MODIS-DT a distinctive version designed specifically for assimilation purposes was used, which was corrected using as a basis four years of collocated data with AERONET (Hyer et al., 2011; Shi et al., 2011; Zhang and Reid, 2006). It is noted that both MODIS-DT and MODIS-DB come with their own uncertainties, thus the comparison with the
assimilated experiments cannot be considered a validation within the strict definition of the term. The uncertainties of these products is discussed in Di Tomaso et al. (2017).

### 2.2 Model simulations

The atmospheric global coupled climate-aerosol modeling system ECHAM-HAM is used as the forward model to generate an ensemble of short-term forecasts. ECHAM is the general circulation part of the modeling system and it simulates the
meteorological conditions of the atmosphere in a Gaussian grid, while HAM is the aerosol module that utilizes the meteorological (e.g. wind, turbulence, convection, precipitation) and surface (e.g. surface roughness, bare and vegetated surface fraction) variables of ECHAM to solve the physical and chemical aerosol particle processes.

### 2.2.1 The ECHAM6-HAM2 Aerosol Climate Model

This study uses the sixth generation of the general circulation model ECHAM6.3, which was developed at the Max Planck
Institute for Meteorology (MPI-M) in Hamburg, Germany (Stevens et al., 2013). The adiabatic processes in the model are based on a spectral-transform dynamical core that simulates some essential meteorological parameters (temperature, surface pressure, vorticity and divergence) while a collection of physical schemes parameterizes the  diabatic processes like convection, diffusion, turbulence and gravity waves (Schultz et al., 2018).

We employ ECHAM-HAM with a grid-resolution of T63L47 (1.875° × 1.875°, with 47 mostly tropospheric levels based on
a hybrid-sigma coordinate). The radiative transfer calculation  in the model are performed by the Rapid Radiation Transfer Model for Global modeling (RRTM-G; Iacono et al., 2008). ECHAM can optionally force the essential meteorological parameters of its adiabatic dynamical core to approach a prescribed field by applying a relaxation technique with time varying weights (Schultz et al., 2018). Typically, the prescribed field consist of a reanalysis database, like ERA-Interim. It is noted that the physics of the model are not directly influenced by the external nudging data, therefore ECHAM is still the
main driver of the dynamics that are just "nudged" towards a prescribed trajectory that describes the 3D temperature, surface


pressure, vorticity and divergence (Rast et al., 2015). Nudging time scales are 24 hours for temperature and surface pressure, 48 hours for divergence and 6 hours for vorticity.

The Hamburg Aerosol Model (HAM) simulates the physical and chemical processes of aerosol in the atmosphere (Stier et al., 2005; Zhang et al., 2012). The most recent version HAM2.3 includes new emission schemes for aerosol and aerosol precursors and modified aerosol-cloud interactions, that are summarized in Tegen et al. (2019). The M7 aerosol model used in HAM2.3 considers five groups of aerosol species, Desert Dust (DU), Sea Salt (SS), Organic Carbon (OC), Black Carbon (BC) and Sulphates ($SO_4$) (Vignati et al., 2004). Nitrate aerosol particles ($NO_3$), that may be produced by gas-phase nitrate ($HNO_3$) and ammonia ($NH_3$) reaction is not implemented in HAM2.3. Aerosols are partitioned in seven unimodal lognormal particle size distributions, called modes, separated into two hygroscopic classes (hydrophobic and hydrophilic). Six of these modes consist of an internal mix of various aerosol types. For the Nucleation mode radius r < 0.005μm, for the Aitken mode 0.005μm < r < 0.05μm, for the Accumulation mode spans 0.05μm < r < 0.5μm and for the Coarse mode is r > 0.5μm (Vignati et al., 2004). The cloud and aerosol optical properties are computed using Mie Theory for each band of the RRTM-G and organized in lookup tables (Tegen et al., 2019). Absorption and scattering of aerosol particles in the ECHAM-HAM is calculated using the prognostic concentrations of aerosol tracers (Schultz et al., 2018).

All aerosol species are emitted, transported and deposited, while depending on their physical and chemical properties they can take part in a number of other processes like aerosol-radiation interactions (scattering and absorption) as well as other aerosol microphysical processes (e.g. nucleation, coagulation, aerosol water uptake and cloud activation). The purely natural emitted aerosol types (DU, SS) are introduced to the atmosphere by utilizing the simulated information of ECHAM, mainly the wind and some surface and ocean characteristics. Aerosols that can be emitted or formed chemically by both natural and anthropogenic sources (OC, BC, $SO_4$) are introduced using predefined emission inventories (Zhang et al., 2012).

Sea salt emissions are parameterized using the wind velocity at 10m as the dominant driver for aerosol particle production, while sea surface temperature (SST) influences mostly the emitted particles size (Long et al., 2011; Sofiev et al., 2011). In cases where the SST is low, sea salt emissions are lower and the emitted particles are smaller compared to when SST is higher (Sofiev et al., 2011). Sea salt particles are emitted only in the soluble accumulation and coarse mode. Natural emissions of dimethyl sulfide (DMS) over the ocean, which is an aerosol precursor, are calculated online based on the 10m wind velocity (Nightingale et al., 2000) and the prescribed concentration of DMS on the surface of the ocean (Lana et al., 2011).

Dust emission are based on the dust source scheme developed by Tegen et al. (2002). Improvements were made in terms of the surface aerodynamic roughness length, soil moisture and soil properties specifically over East Asia by Cheng et al. (2008). Also there were updates regarding the representation of Saharan dust sources using infrared dust index from the SEVIRI instrument upon Meteosat Second Generation Satellite by Heinold et al. (2016). Wind velocity at 10m is the main driver of dust aerosol particle production while soil properties are taken into account. Saltation processes are simulated following (Marticorena and Bergametti, 1995). The surface roughness length is fixed globally to the value 0.001 and the minimum friction velocity threshold for dust mobilization is set to 21 cm s$^{-1}$. The threshold friction velocity depends on the





soil size distribution, vegetation cover and soil moisture (Cheng et al., 2008). The preferential dust emission sources include arid or low vegetated areas and are predefined in accordance to Tegen et al. (2002). Dust particles are initially emitted in the insoluble accumulation and coarse mode, but aging processes like the condensation of sulfuric acid ($H_2SO_4$) onto insoluble dust particles leads to soluble dust in the accumulation and coarse modes (Zhang et al., 2012).

The emission for the remaining aerosol types and aerosol precursors are defined using emission inventories organized in 14
sectors, with each sector corresponding to the emission flux of one or more aerosol types or aerosol precursors (Schultz et al., 2018; Tegen et al., 2019). The Atmospheric Chemistry and Climate Model Intercomparison Project (ACCMIP) dataset has been used for the anthropogenic aerosol and aerosol precursor emissions, which consists of monthly mean estimates at a horizontal resolution 0.5°x0.5° (Lamarque et al., 2010). The first version of Global Fire Assimilation System (GFAS) was employed for the representation of biomass burning emissions coming from grass and forest fires. GFAS is a gridded daily
product with 0.5°x0.5° horizontal resolution based on the fire radiative power measurements of MODIS instrument (Kaiser et al., 2012). The daily temporal resolution of GFAS and its accurate spatial representation of fires is the critical characteristics that makes it ideal for the daily assimilation cycle applied in this study. Using standard GFAS emissions, several studies reported underestimated AOD, thus a fire emission factor equal to 3.4 has been proposed (Kaiser et al., 2012; Tegen et al., 2019; Veira et al., 2015). This factor only correct for the AOD bias and not the AAOD bias, and therefore the
GFAS emission rescaling factor 3.4 has not been adopted in the present study.

Most species are emitted at the lowest level of the model which represents the surface of the Earth. Aerosols related to energy production and ships are emitted directly to the second lowest level of the model. In HAM2.3 the 75% of biomass burning emissions is equally distributed in the PBL, while 17% and 8% are emitted in the first and second level above the PBL respectively (Val Martin et al. (2010) reported that most BB emissions occur within PBL). Although it is noted that the
wildfire emission height has a limited effect on global AOD distribution when compared to emission fluxes and dry/wet deposition processes (Veira et al., 2015). The partitioning of aerosol emission to the M7 modes is described in detail in Schutgens and Stier (2014). It is noted that all the experiments of this study use rescaled emissions based on the analysis of subsection 0.

## 3. Methods

**3.1 Local Ensemble Transform Kalman Filter (LETKF)**

In this study we use the Local Ensemble Transform Kalman Filter (LETKF). The Kalman equation (Rodgers, 2000) involves the forecast, also called a-prior or background state ($x_b$) of the system, the analysis, also called a-posterior or assimilated state ($x_a$) of the system, the observational data ($y$), the observational operator ($H$) and the two error covariance matrices that describe the uncertainties and correlations in the background estimates ($P$) and the observation estimates ($R$):

$$x_a = x_b + G \cdot (y - H \cdot x_b) \qquad (1)$$



$G = P_a \cdot H^T \cdot R^{-1}$ is called the Kalman Gain. The subscripts a and b indicate the analysis and background states respectively and T denotes the transpose operator. Equation (1) states that the analysis state ($x_a$) is computed based on the background state ($x_b$) plus the product of the Kalman Gain and the difference between the observations and the simulated observations of the background state ($y - H \cdot x_b$), called innovation. Kalman Gain is a matrix of weights, which corresponds to every ensemble member and adjusts the impact of innovation on the new analysis state. If Kalman Gain is equal to zero, this means that either the model covariance is zero, the observational covariance error matrices is infinite, or there is no dependence of the measurements on the state vector elements, which implies that observations hold no useful information and $x_a$ is set equal to $x_b$. The analysis covariance error ($P_a$) can be calculated from the background state of the ensemble:

$$P_a = P_b \cdot (I + P_b \cdot H^T \cdot R^{-1} \cdot H) \tag{2}$$

where **I** is the identity matrix. Equation (1) and (2) can be also expressed as the minimization of the cost function:

$$\Psi(x_a) = (x_b - x_a)^T P^{-1} (x_b - x_a) + (y - H \cdot x_a)^T R^{-1} (y - H \cdot x_a) \tag{3}$$

where the two components at the right side of the equation describe the difference between background and analysis state and the difference between observations and analysis state. In a nutshell, the minimization of cost function (3) defines a new better estimate of the state vector based on observational data, a background estimate and by taking into account the errors in both the model and observations.

The comprehensive mathematical formulation of LETKF can be found at Hunt et al. (2007). LETKF was implemented by Schutgens et al. (2010a) for an aerosol application based on previous work by Miyoshi and Yamane (2007). The code was modified to operate with ECHAM-HAM and POLDER in this paper.

The state vector of the system (**x**), is in our case composed of simulated ECHAM-HAM aerosol mass mixing ratios for every mode and species and number mixing ratio for every mode (23 in total; Table 1).

The observations vector **y** consist of aerosol optical properties retrieved by POLDER. The observation operator **H** transforms the aerosol mixing ratio into simulated aerosol optical properties and uses the optical properties routines in ECHAM-HAM. The perturbed ensemble of ECHAM-HAM (subsection 3.2) embodies the model error covariance matrix **P**, while the calculated POLDER errors (subsection 2.1.2) are utilized in the observation error covariance matrix **R**. **R** assumes uncorrelated between the assimilated variables, hence the off-diagonal elements of the matrix are set to zero. In reality correlations do exist between variables and thus it can affect the assimilation results. The off-diagonal elements of the **R** matrix can be estimated by constructing a data-derived **R** matrix (Liu et al., 2019), but it is out of the scope of the present study.

## 3.2 Model Uncertainties and Ensemble Perturbation

When creating the ensemble to represent the model error covariance matrix, we consider 2 sources of uncertainty, namely in the aerosol emissions and in the wind-speed/direction. Global climate models or climate transport models estimates of aerosol concentration and aerosol optical properties in the atmosphere are diverse and uncertain. The most prominent cause of these uncertainties mainly originates from the emission of natural aerosols, which depending on the type, global estimates




may differ between 4 to 16 times (Grythe et al., 2014; Huneeus et al., 2011; Lewis and Schwartz, 2004; Miller et al., 2006;
Pan et al., 2019; Textor et al., 2006). A recent study highlighted that fire emission inventories for OC and BC total global
emission of 6 biomass burning emission inventories differ by a factor of 4 (Pan et al., 2019). Furthermore, a multi-model
study from AEROCOM phase-I (Huneeus et al., 2011) and single-model studies using different DU emission schemes
(Miller et al., 2006) indicate that DU global emission fluxes may differ by up to a factor 6. SS global emission fluxes show
the highest uncertainties among all other aerosol species mostly due to the differences on the simulated particles size (Textor
et al., 2006) and the differences in the sea-spray aerosol function of the models (Grythe et al., 2014). A wide range of SS
fluxes have been reported depending on the sea-spray aerosol function used (Grythe et al., 2014), but the most well accepted
range for SS emission flux is between 1.2-20 Pg·yr$^{-1}$ (Lewis and Schwartz, 2004), which implies an approximately 16 times
SS emission flux difference between the highest and the lowest estimates.

Contrary to natural aerosol emissions, the anthropogenic aerosol emissions (OC, BC, SO$_4$) and its precursors (SO$_2$) are better
constrained. For eastern China the diversity (highest to lowest emission inventory) is lower than 1.35 for OC, BC and SO$_2$
(Chang et al., 2015), although for other regions like South America diversity was estimated up to 3 (Granier et al., 2011). It
is noted that most anthropogenic emission inventories are on a monthly basis, hence the day to day variability is not
accounted in model simulations. It stands to reason that the uncertainty in daily anthropogenic emissions should be higher
than the aforementioned values derived for monthly emissions. Other studies have noted that deposition parameterization
cause large uncertainties in the direct and indirect aerosol radiative effect (Lee et al., 2016; Regayre et al., 2018), hence may
be considered in our future studies.

In addition to the emission uncertainties, meteorological factors may affect the emission, transport, deposition and hence the
atmospheric lifetime and radiative effect of aerosol particles. In our simulations, the surface pressure vorticity and
divergence of ECHAM is nudged towards the ERA-interim reanalysis. Although ERA-interim is a reanalysis product that
provides a better estimate of the meteorological conditions, it contains uncertainties that may affect the life cycle of aerosol
particles. Several studies using wind measurements from ground station, radiosondes, buoys, cruises ships or even satellite
estimations revealed ERA-interim errors by up to ±3 m·s$^{-1}$ (Bao and Zhang, 2013; Bromwich et al., 2016; Brunke et al.,
2011; Campos and Guedes Soares, 2017; Stopa and Cheung, 2014).

Considering these sources of aerosol uncertainty, originating from aerosol emissions and wind speed/direction our ECHAM-
HAM forecast ensemble was assembled by multiplying the standard aerosol emissions with spatially correlated perturbations
to obtain the emission for each member and nudging each member to a slightly different version of ERA-interim reanalysis
in terms of wind speed.

A typical approach for the emission perturbation of the ensemble in past studies is to either use global emission perturbations
unique for each member and constant through time (Dai et al., 2014; Schutgens et al., 2010a). Another approach is to use
spatiotemporal independent emission perturbations where random numbers are assigned for each grid cell and often these
numbers change in time. The latter case proved to produce very low spread in the ensemble, making the assimilation of
observations impractical (Dai et al., 2014; Schutgens et al., 2010a). An intermediate approach is adopted in the present study,





by using spatially correlated emission perturbations, where the changes from grid to grid are not abrupt but smooth, perturbing emissions with positive or negative numbers over large areas. This technique has been successfully used in the

past to derive soil erodibility factors under Observation System Simulation Experiments (OSSEs) using an ensemble adjustment Kalman Filter (Khade et al., 2013).

Spatially correlated perturbations fields are generated by firstly creating an ensemble (32 members in our standard setup) global grid at the resolution of the model (1.875°x1.875°) filled with random values, sampled from a gaussian distribution. This spatial field of perturbations does not contain any spatial correlation between the neighbouring grid cells (the spatial

field for one member is shown in FigureS 1a). In the next step, the global grid is smoothed by averaging, for each grid box, the surrounding grid boxes that are within a 2 grid box distance (FigureS 1b). The last step is repeated 5 times in total (FigureS 1b-f). Next an exponential function is applied for each grid cell separately making the values positive and the distribution positively skewed (close to log-normal) (FigureS 1g). The final step standardizes the numbers (V) to a mean ($MV_{NEW}$) equal to the rescaled factors at FigureS 5 (more information at subsection 3.4) and a standard deviation ($SV_{NEW}$)

equal to 0.65 for each grid (FigureS 1h), in the form of:

$$V_{NEW} = SV_{NEW} \cdot \frac{V-MV}{SV} + MV_{NEW} \qquad (4)$$

The variogram model for the spatial field of a member for each of the steps is shown in FigureS 2. According to it, in the final step (FigureS 2h) the variogram model flattens at the distance of 30°, which indicates that grid cell are spatially correlated up to the distance of 30°, whereas locations further than that are not.

The aforementioned methodology produces 32 different spatially correlated maps, one for each ensemble member. Each aerosol species has its own unique randomly generated set of 32 spatially correlated maps. The aerosol emission fluxes of the model are multiplied (perturbed) with these maps while the model runs. The standard deviation of these spatially correlated perturbations (hence the uncertainty of the emissions) for each species is equal to 0.65. This simplified approach assumes that the natural and anthropogenic emissions will have the same level of uncertainty.

Similar steps were followed for the creation of the zonal and meridional components of the wind spatially correlated perturbations. Contrary to the emission fluxes where numbers should be strictly positive, the wind vectors sign indicates direction and the values can be negative, thus the exponential function was not applied. The values were standardized with a mean and a standard deviation of 0 and 0.8 respectively using Eq. 4. The final wind perturbations are a selection of 32 members, different for the two wind components. These perturbations are added to the wind fields of ERA-interim reanalysis

dataset, creating different perturbed branches that each member of the ensemble is nudged to. That approach can account partially for the uncertainties on the wind. A variety of other wind perturbation methods were tested, like altering the nudging relaxation time or restarting the model from different initially conditions, but were not adopted in the final experiments because either the produced ensemble spread was too small or it shrank close to zero after some days of simulation.





The distribution of perturbations for the dust emission and U-component of the wind is shown in FigureS 3. An example of the emission and wind spatially correlated perturbations maps generated for a single member is shown in Figure 2. In this example, the DU emission of this member are higher than the default emission parameterization over the Arabian Peninsula and the Central-Eastern Sahara, while the western part of the Sahara will have lower DU emissions (Figure 2a). Similarly the U-component (positive eastward, negative westward) of the nudging data for wind over the desert will be 1 m·s$^{-1}$ higher,

which close to the surface may increase the emission of dust particles but higher up can also reduce the westward outflow of dust towards the North Atlantic (Figure 2b). Other members may have a different combination of these parameters in the same area, which may cover other possible scenarios that may match reality.

The derived model uncertainty, shown in the Figure 3 as the standard deviation of this perturbed ensemble, can be conceptually compared with the POLDER uncertainty (Figure 1). In all cases the global mean of model uncertainty is lower

than the POLDER uncertainty. This does not mean that the simulated aerosol observations of the model are more accurate in comparison to POLDER. The global mean uncertainty of the model is lower since low aerosol regions (majority of points globally) are far away from emission sources. Hence, the emission perturbation has a very limited effect and the members within the ensemble are quite similar with each other in the low aerosol regions. Moreover, the global mean of the model includes polar regions where the absolute uncertainty is very low. In the remote regions, POLDER microphysical retrievals

are uncertain in an absolute sense and POLDER AOD retrievals are uncertain in a relative sense. On the other hand, the exact opposite is observed close to emissions sources. For example the $SSA_{550}$ model uncertainty over the southern part of Africa is up to 0.1 (Figure 3h) while for the same region the POLDER uncertainty is only ~0.04 (Figure 1h). A notable difference between the model and POLDER uncertainties can be also observed for $AOD_{550}$ in any outflow region over the ocean (e.g. North Atlantic Ocean, South Atlantic Ocean, South Indian Ocean) (Figure 1a and Figure 3a). Furthermore, it is

noted also that other uncertainty factors related to aerosol physical and chemical processes (deposition, aging, vertical convective transport) are not taken into account in our model uncertainty estimation.

### 3.3 Data Assimilation System

Our system consists of three phases, namely the spin up, the perturbation and the data assimilation phase. The first phase, includes a single simulation that runs from January 2006 till the end of May 2006 and is a spin up simulation for the

ECHAM-HAM meteorological and aerosol state. The second phase consists of an ensemble of runs from June to 28$^{th}$ of August 2006, where each member's aerosol emission (DU, SS, OC, BC and $SO_4$) and wind (U-Zonal and V-Meridional component) are perturbed using random spatially correlated maps (Section 0)., which serves as the representation of model error in the assimilation phase. The third stage is the assimilation of observations, solved in a daily cycle from 20$^{th}$ of July until 28$^{th}$ of August 2006 (40 days).

The daily cycle of data assimilation involves daily forecasts of all perturbed ensemble members. Upon completion of these simulations, the LETKF code is called which performs a spatial collocation of the simulated (ECHAM-HAM) and the retrieved (POLDER) observations for four temporal time-steps (00, 06, 12, 18 UTC). Subsequently LETKF computes a new



analysis state vector (ECHAM-HAM aerosol mixing ratio) at the last time step of the day, which will serve as initial conditions for the next day's forecast. The process is repeated till the end of the data assimilation experiment.

At the grid cell scale, LETKF inflates the observation errors depending on their distance from the assimilated grid. This method, known in literature as "observation localization", aims to reduce error covariance between distant points which is caused by sampling errors due to the limited ensemble size (Miyoshi and Yamane, 2007). More specifically the local patch size ($L_x$) defines the distance between the analyzed grid cell and the observations that will be taken into account for assimilation. The observational error (E) of the observations that are within $L_x$, are adjusted ($E_A$) according to their distance

(D) in grid cell units from the assimilated grid using a horizontal correlation length ($L_y$):

$$E_A = E \cdot \exp\left(D/L_y{}^2\right) \tag{5}$$

Distant observations get higher errors, thus having a smaller contribution in the changes of the assimilated grid. In our experiments $L_x$ and $L_y$ are set to 4 and 2 respectively in grid cell units. Consequently, observations that are up to 4 grids (7.5°) away from the assimilated grid may affect the assimilated grid cell, although these more distant observations are

accounted with a 2.7 greater observational error than normal. An illustration of the daily assimilation cycle and the grid cell scale collocation during assimilation is depicted at Figure 4.

### 3.4 Rescaling the Aerosol Emissions

Aerosol models often may under-or overestimate $AOD_{550}$ or $AAOD_{550}$ close to the sources. Most of these biases close to sources may be attributed to inaccurate emissions of one or two aerosol species. By rescaling these emissions by region and

species the model simulated observations will be closer to the real observations. This rescaling of emissions can benefit the data assimilation since the Kalman filter assumes that the model is unbiased. Therefore, a simulation for 2006 was conducted with ECHAM-HAM in order to identify yearly $AOD_{550}$ and $AAOD_{550}$ biases by evaluating it against POLDER and MODIS-DT retrievals.

Initially, the mean yearly bias of the 2006 simulation against MODIS-DT $AOD_{550}$, POLDER $AOD_{550}$ and POLDER

$AAOD_{550}$ and the yearly emission fluxes for all species were plotted (FigureS 4). The results indicate that a positive bias of the model against MODIS-DT $AOD_{550}$ is most probably driven by an overestimation of SS emission fluxes, since these two spatial patterns match (FigureS 4a,e). For the same reason, the over/underestimation of the model when compared to POLDER $AAOD_{550}$ over wildfire or anthropogenic polluted regions may be mainly attributed to BC emissions fluxes (FigureS 4c,f). Furthermore the biases of POLDER $AOD_{550}$ over desert is for the most part associated with DU emissions

fluxes (FigureS 4b,d). Emission rescaling factors (RF) for $SO_2$, $SO_4$, SS and OC were based on biases against MODIS $AOD_{550}$, DU based on biases against POLDER $AOD_{550}$ and BC based on biases against POLDER $AAOD_{550}$ (FigureS 5). The emission RF were calculated as the ratio of observations (OBS) to model (MOD) $AOD_{550}$ or $AAOD_{550}$ for each region (r):

$$RF_r = \frac{OBS_r}{MOD_r} \tag{6}$$



Afterwards the global grid was smoothed to avoid spatially steep changes of aerosol emission fluxes, by averaging each grid
cell with the surrounding values at a distance of 2 grid cells. This approach does not account for observation errors, it does
not consider the inter-annual and intra-annual biases and it is noted that the aerosol emission adjustment is based on $AOD_{550}$
and $AAOD_{550}$ biases, which may not be directly related to aerosol emissions.

An obvious limitation of this method occurs over the southern hemisphere major fire sources, where $AOD_{550}$ is
underestimated and $AAOD_{550}$ is overestimated by ECHAM-HAM. Following the above-mentioned methodology, the
resulting emission factors for fire source regions of the southern hemisphere would result to a reduction of BC emission
fluxes by 10%, due to the underestimation of $AAOD_{550}$, and an increase of OC emission fluxes by more than 50%, due to the
underestimation of $AOD_{550}$. The outcome would provide an improvement in terms of $AOD_{550}$ but not for $AAOD_{550}$, since
part of the $AAOD_{550}$ (10%-20%) emerges from other species, like the OC. Thus, in order to get a simultaneous improvement
in both $AOD_{550}$ and $AAOD_{550}$, OC emission fluxes were not adjusted in wildfire regions in the Southern hemisphere major
fire regions (FigureS 5c). The emission perturbation of the ensemble that is used in the core-experiments are up to 3 times
greater than the rescaling factors applied in this stage, thus the high and low values of the original unadjusted yearly
simulation are still represented in the perturbed ensemble.

## 3.5 Experimental Setup

The experiments are focused on summer of 2006 and assimilation is performed for the period 20[th] of July to 28[th] of August
2006. The year was selected according to the availability of POLDER SRON retrievals, while the summer season was
chosen due to the high peak of forest fires in the tropical band and the pronounced dust plume over the Atlantic, where
AAOD and AE in the model can benefit from the assimilation process. The model simulations were bilinearly interpolated
when necessary to a $1° \times 1°$ global grid for a direct comparison with the gridded satellite retrievals.

The core-experiments presented in Section 4.1, assess the potential added value of assimilating aerosol information related to
the size and absorption. In the CONTROL experiment, there was no assimilation of any kind of observations. In the MASS
experiment, only $AOD_{550}$ was assimilated. In the experiments SIZE1 and SIZE2 either $AOD_{550}$ & $AOD_{865}$ or $AOD_{550}$ &
$AE_{550-865}$ were assimilated, respectively. In the experiments ABSORB1 and ABSORB2, either $AOD_{550}$ & $AAOD_{550}$ or
$AOD_{550}$ & $SSA_{550}$ were assimilated, respectively. Finally, the experiment TOTAL assimilates $AOD_{550}$ & $AE_{550-865}$ & $SSA_{550}$.
Sensitivity-experiments described in subsection 4.3, intend to explore LETKF's sensitivity to three main parameters,
ensemble size ($n_{ens}$), local patch size ($L_x$) along with horizontal localization factor ($L_y$) and inflation factor ($ρ$) following an
analogous analysis conducted by (Schutgens et al., 2010b). The $n_{ens}$ essentially is the number of members used in ensemble
and is connected to the accuracy and diversity of the model predicted covariant error. The $L_x$ represents the distance around a
model grid that defines whether an observation would be considered in the assimilation process, while $ρ$ relates to a
technique that multiplies the error covariance matrix of the ensemble to increase the ensemble spread and ensure the
assimilation of new observations is possible in the next assimilation step (because otherwise the ensemble spread decreases
during the assimilation process). Table 2 presents a summary of the core and the sensitivity experiments.

## 4. Results

### 4.1 Comparing to POLDER observations

We first evaluate the impact of data assimilation by evaluating the daily forecast, started from the latest analysis, with
POLDER data not yet assimilated. The benefit of this sort of evaluation is that biases in POLDER retrievals are effectively
removed (because the observations used for either assimilating and evaluation come from the same POLDER dataset) and
one can study the merits of data assimilation without the added issue of observational biases. The drawback is that such an
evaluation cannot determine if the assimilation of POLDER retrievals actually yields improved aerosol simulation. The
evaluation of the system with completely independent observations is presented in subsection 0. Each experiment consists of
an ensemble of simulations. The ensemble mean of each experiment is the best estimate for the state vector and the
simulated observations. Thus, only the ensemble mean of each experiment is presented in the results. Throughout the results
we use the forecast run of the most recent analysis.

### 4.1.1 Aerosol Optical Depth

Figure 5 and Figure 6 shows the $AOD_{550}$ fields for the different data assimilation experiments of Table 2 and their agreement
with the POLDER $AOD_{550}$. Looking at the CONTROL experiment, we see that the model underestimates $AOD_{550}$ over land
in most regions with high $AOD_{550}$, such as the Sahara (Dust), tropical Africa (biomass burning), and Asia (industrial). On the
other hand, over ocean the model has the tendency to overestimate $AOD_{550}$. Overall, the global mean $AOD_{550}$ of the model
(0.146) is substantially lower than that of POLDER (0.228).

When assimilating $AOD_{550}$ (experiment MASS) the global bias virtually disappears, which indicates the data assimilation
system is well capable of using the AOD information provided by POLDER measurements. When more properties than just
$AOD_{550}$ are assimilated (SIZE2, ABSORB2, TOTAL), the bias in $AOD_{550}$ gets a bit larger than when only assimilating
$AOD_{550}$ (-0.013 to -0.020) but still much smaller than for the CONTROL experiment. The reason that the agreement with
POLDER $AOD_{550}$ gets slightly worse when assimilating other properties in addition to $AOD_{550}$ (AE and/or SSA), is that the
system needs to find the best compromise in fitting all properties simultaneously. So, in some situations the only way to get a
better fit to (e.g.) AE is to degrade the fit to $AOD_{550}$ (because of different assumptions used in the model). On the other hand,
it is very important to note that the assimilation of other aerosol optical properties like $AE_{550-865}$ and $SSA_{550}$ reduced the
South Atlantic positive $AOD_{550}$ bias after assimilation, especially in the case of the TOTAL experiment (Figure 5i),
indicating that the simultaneous assimilation of multiple variable enhances the simulated $AOD_{550}$ global spatial
representation.

The $AOD_{550}$ scatterplots for all core-experiments are depicted Figure 6. The averaged global Mean Error (ME) is reduced
from -0.071 in the CONTROL to values that range between -0.002 to 0.020 in the assimilated experiments. Similarly, the
Mean Absolute Error (MAE) is reduced from 0.109 in the CONTROL to 0.078 in the TOTAL experiment. Pearson's
Correlation (R) increases from 0.668 to approximately 0.8 for all assimilated experiments. The consistent improvement of





AOD$_{550}$ in the assimilation experiments demonstrates the ability of the assimilation system to adjust aerosol mixing ratio regardless of combination of assimilated observations.

### 4.1.2 Angstrom Exponent

Figure 7 and Figure 8 depicts AE$_{550-865}$ for POLDER and the data assimilation experiments. The CONTROL experiment overestimates the AE$_{550-865}$ in most cases (Pacific and Indian Ocean, Australia, Siberia) and underestimates it at the western Sahara. Globally the mean AE$_{550-865}$ of CONTROL (1.25) is higher than that of POLDER (0.95), which indicates that the model overestimates the ratio of fine to coarse mode particles.

The MASS experiment has lower global AE$_{550-865}$ (1.12), which matches slightly better, but still being higher than POLDER. In the MASS experiment only AOD$_{550}$ is assimilated, thus any information regarding size or chemical composition will be related to a combination of transport and previous cycles that assimilated AOD$_{550}$ close to sources. More specifically, particle size information may indirectly be introduced into the assimilation system by adjusting the AOD of two neighbouring regions with different dominant particle size distribution, like North Africa and Europe. For example, in Europe the spatial averaged bias of AE$_{550-865}$ in the CONTROL experiment was 0.20 while in MASS it is 0.01 (FigureS 6).

The other experiments (SIZE2 ABSORB2 and TOTAL), reduce the global mean of AE$_{550-865}$ even more (1.06, 1.16 and 0.98 respectively). The combined information of more observations related to mass, size and absorption reduces the local biases of AE$_{550-865}$ as well as the global ME and MAE while increasing R (Figure 8g). Also, in the TOTAL experiment the AE$_{550-865}$ is improved substantially even in areas where the POLDER uncertainty of AE$_{550-865}$ is quite high. For example, the POLDER uncertainty in Australia is approximately 0.5 (Figure 1g), but CONTROL overestimates AE$_{550-865}$ by 0.8 over land (Figure 7g).

### 4.1.3 Aerosol Absorption

Figure 9 and Figure 10 show the agreement in AAOD and SSA, respectively, between the different experiments and the POLDER observations. The CONTROL experiment shows lower global mean aerosol absorption than the POLDER observations, i.e. the AAOD is lower and the SSA is higher. Regionally, the differences are more complex. For example, over the southern part of Africa, with a lot of biomass burning, the AAOD is low compared to POLDER as well as the SSA. This means that the difference in AAOD is mostly caused by too low total AOD, but that the aerosols are more absorbing "per particle" in the CONTROL experiment than in POLDER. A similar pattern is observed over northern Eurasia. On the other hand, over the middle east and over North America, a low bias in AAOD is observed together with a high bias is SSA, while for South America a high bias in AAOD is observed together with a low bias in SSA. So, in these regions the differences between POLDER and the CONTROL experiment are caused by differences in absorption properties of the aerosols.

It is interesting to note that the assimilation of only POLDER AOD$_{550}$ (MASS experiment) improves the total AOD$_{550}$ representation of ECHAM-HAM (Figure 5h), but it negatively affects the spatial representation of the SSA$_{550}$ and AAOD$_{550}$



in the model in key areas like the South America, Africa and the Atlantic Ocean (Figure 9h, Figure 10h). The reason behind that is easiest to explain over South America, where $AOD_{550}$ is underestimated and AAOD (SSA) is overestimated (underestimated) in the model. Hence, the assimilation of POLDER $AOD_{550}$, will correct (increase) for the total extinction of BC, which is already overestimated in the CONTROL run, but not for its absorption. Specifically, in Amazon basin $SSA_{550}$

of the MASS experiment decreases by 0.032 in comparison to CONTROL, since the BC column burden becomes 4 times higher (FigureS 7b), while the difference of $SSA_{550}$ between POLDER and the model (spatiotemporal collocated points only) increases from -0.084 to -0.117 (FigureS 7c).

Scatterplots of all experiments for $AAOD_{550}$ and $SSA_{550}$ are depicted in Figure 11 and Figure 12 respectively. The global $SSA_{550}$ ME gets worse from 0.013 to -0.025 for the MASS experiments, the MAE increases also from 0.049 to 0.058, while

R decreases from 0.243 to 0.162 (Figure 12a,b). These results reveal the limitations of an aerosol data assimilation system that assimilates AOD in only one wavelength.  Assimilating size information does not really improve the agreement in AAOD or SSA, but as expected, when assimilating SSA (ABSORB2 experiment), the agreement between model and POLDER data significantly improves for both AAOD and SSA. This improvement is maintained for the TOTAL experiment, assimilating AOD, AE, and SSA together. The $SSA_{550}$ of ABSORB2 experiment is still slightly higher over the

Amazon basin and lower in the Middle East (Figure 10k) but overall the simulated absorbing properties are significantly better in comparison to the CONTROL experiment (Figure 10g) and much better in comparison to MASS experiment (Figure 10h).

### 4.1.4 Aerosol column burden changes

The aerosol column burden changes for each experiment are depicted in Table 3. All experiments reveal that ECHAM-HAM

underestimates aerosol column burden globally, thus the changes for all species are positive. In the experiment MASS, BC column burden is almost 5 times greater than the CONTROL experiment. BC $AOD_{550}$ contribution to $AOD_{550}$ is less than 10% in most regions over the globe, thus the assimilation of only $AOD_{550}$ shouldn't affect the BC column burden significantly. On the other hand, OC $AOD_{550}$ contribution to $AOD_{550}$ is between 50-90% in the tropical fire and outflow areas and thus the assimilation of $AOD_{550}$ is expected to affect significantly the OC column burden. Although BC and OC

emissions are perturbed differently, correlations in these two species will still persist, since both BC and OC are emitted from the same location (but not with the same magnitude) and are following similar transport paths in each member. Thus, concluding that the large BC column burden increase is related also to the correlations between OC and BC $AOD_{550}$. The experiment ABSORB1 constrains the BC increase to +27% in comparison to CONTROL. The experiment TOTAL increases of the column burden ranges between +20% to +95% for all the species. To understand better how these changes are made,

we explore further regionally by isolating the effect of assimilating $AE_{550-865}$ and $AAOD_{550}$ for Australia and South America respectively.

Figure 13 depicts the aerosol optical properties and aerosol column burden changes over Australia caused by the assimilation of $AE_{550-865}$ by subtracting the value for the experiment MASS from SIZE2 (SIZE2-MASS). Australia is a mix aerosol area



with fairly low aerosol content and many uncertain emission sources within the continent, while satellite retrievals are quite
diverse (Schutgens et al., 2020) due to the complex surface albedo of the continent. MASS clearly overestimates the amount
of fine particles over the Australian continent compared to POLDER (Figure 7h), hence the addition of $AE_{550-865}$ in the
assimilation increases the column burden on rather coarser aerosol groups (DU and SS) and decreases the column burden the
modes corresponding to fine particles, where OC, BC, and $SO_4$ are dominant. (Figure 13c). Consequently, the $AE_{550-865}$ bias
against POLDER is reduced from 0.282 in the MASS experiments to 0.166 in the SIZE2 experiment. The bias is decreased
also in some other parameters, like $AOD_{550}$, $AAOD_{550}$ and $SSA_{550}$. It is noted that the aerosol mixing ratio changes on the
assimilation system are conducted for every aerosol tracer separately (Table 1), but since some aerosol groups contain higher
column burden in coarser aerosol modes (DU and SS) and other in finer aerosol modes (OC, BC, $SO_4$), aerosol adjustment
due to assimilation are regularly consistent (either positive or negative) for an aerosol species as a whole, like in the case of
Australia.

Similarly, Figure 14 depicts the aerosol optical properties and aerosol column burden changes over South America caused by
the assimilation of $AAOD_{550}$ by subtracting the experiment MASS from ABSORB1 (ABSORB1-MASS). South America
and specifically the Amazon basin, is a major active burning area of the globe in July and August with significant emissions
of absorbing aerosol particles. When assimilating $AOD_{550}$ & $AAOD_{550}$ (ABSORB1) the absorption optical properties are
improved (Figure 14c), while $AOD_{550}$, $AOD_{865}$ and $AE_{550-865}$ performance slightly deteriorates. The changes in the simulated
absorbing properties are mainly driven by a -81% reduction of the BC mixing ratio, which reduces the $AAOD_{550}$ by -0.024
and increases the $SSA_{550}$ by 0.078 in comparison to the experiment MASS (Figure 14b). DU and OC changes may in the
experiment ABSORB1 affect a small fraction of $AAOD_{550}$ changes too, but predominantly the other species adjust to match
as good as possible the other assimilated parameter ($AOD_{550}$).

## 4.2 Comparing with the independent observations

In this subsection, we evaluate the aerosol fields from the different data assimilation experiments using independent
observations. AERONET is the most important data source for this purpose given its high accuracy, especially for AOD
from its direct sun product. However, spatial coverage by AERONET sites is sparse and entirely absent over the ocean
(except for a few islands and coastal stations). MODIS-DT & DB on the other hand provide close to global coverage and
have been extensively evaluated. Here we present the comparisons of our POLDER assimilation experiments with these
independent datasets.

### 4.2.1 Evaluation with AERONET

In Figure 15 the background simulation (CONTROL) and the total aerosol assimilated experiment (TOTAL) are evaluated
against AERONET Direct-Sun V3 L2 dataset for AOD and AE. The background simulation (CONTROL) shows a clear
negative bias with AERONET in AOD of -0.072. Assimilating POLDER AOD, AE, and SSA simultaneously (TOTAL),
removes this bias almost entirely (remaining bias 0.001). The reduction in MAE is much more moderate (from 0.127 to


0.11). In terms of $AE_{550-865}$, the ME reduce from 0.273 to -0.084 and the MAE reduced from 0.419 to 0.353 (Figure 15 d,e). The spatiotemporal collocated points between POLDER and AERONET indicate that POLDER has a very small ME in $AOD_{550}$ and $AE_{550-865}$ (Figure 15 c,f), thus the assimilation of these variables from POLDER will converge the ensemble mean to a very low bias in comparison to AERONET (Figure 15 b,e).

In Figure 16 the CONTROL, MASS, ABSORB2 and TOTAL experiments are evaluated using the AERONET Aerosol Inversion V3 L2 dataset for $AAOD_{550}$ and $SSA_{550}$. Both properties deteriorate at the MASS experiment, improve in the ABSORB2 experiment (Figure 16 b,e), while in TOTAL experiment only $AAOD_{550}$ improves (Figure 16 c,f). It is noted that the AERONET Aerosol Inversion dataset that provides $AAOD_{550}$ and $SSA_{550}$, contains far fewer stations and hence less collocated points with the model (N=772), in comparison to the AERONET Direct-Sun dataset (N=11832). The 590 spatiotemporal collocated points of POLDER retrievals with AERONET Inversion V3 L2 dataset were very few (N=31) for the study period, thus not shown.

### 4.2.2 Comparing with MODIS-DT and MODIS-DB

The $AOD_{550}$ of the CONTROL and the TOTAL experiments is compared with MODIS-DT and MODIS-DB over land in Figure 17. In both cases the negative ME is reduced in the assimilated experiment from -0.066 to -0.002 (MODIS-DT over 595 land) and -0.103 to -0.029 (MODIS-DB), the MAE is reduced as well and the correlation increases. The assimilation of POLDER observations brings the analysis closer to MODIS-DT and MODIS-DB over land, although the three datasets use different retrievals algorithms. Contrary the comparison with MODIS-DT over ocean reveals that the assimilation slightly increases the ME and MAE (Figure 18 a,b). The reason for this is that the assimilation of POLDER measurements increases the $AOD_{550}$ over land, which may have an effect on the $AOD_{550}$ error over ocean in outflow regions (e.g. South Atlantic; 600 Figure 5l). These results indicate that more observations are needed over the source area of the outflow region (Africa) or more observations over the outflow region (South Atlantic) in order to constrain the ocean $AOD_{550}$. Furthermore, the over ocean $AOD_{550}$ overestimation of the assimilated experiment is more prominent against MODIS-DT, since POLDER has higher global mean AOD than MODIS-DT by 0.032 (Figure 18c).

### 4.3 Sensitivity Experiments

In practice, many assimilation systems assume uncorrelated observational uncertainties, by setting the off-diagonal elements of the observation error covariance matrix **R** to zero. The SIZE2 experiment results in a somewhat better agreement in AE with POLDER than SIZE1 (Figure 8c,d), despite the fact that the observations contain the same information. This is most likely an artefact of existing correlations in the observation uncertainties. The nature of error correlations for these POLDER variables is illustrated in FigureS 8. Pearson's correlation for the first group of variables ($AOD_{550}$, $AOD_{865}$) is 0.92 while for 610 the second groups of variables ($AOD_{550}$, $AE_{550-865}$) is -0.22. When assimilating AOD at two different wavelengths, it becomes important to specify the off-diagonal elements in the **R** matrix. Ignoring so, prevents LETKF from optimally using the information in the observations.



Similar conclusions can be drawn by the experiments ABSORB1 (assimilation of $AOD_{550}$ & $AAOD_{550}$) and ABSORB2 (assimilation of $AOD_{550}$ & $SSA_{550}$). Both experiments are improving the simulated absorbing optical properties of the model
(Figure 11e,f and Figure 12e,f). From the results, it seems that $AAOD_{550}$ is more efficient on reducing the difference to POLDER in high aerosol content situations, like the Amazon basin, where $SSA_{550}$ has larger effect over remote areas with lower aerosol content, like the Pacific Ocean (FigureS 9). Thus, depending on the area of interest, especially in studies that may use a similar assimilation system with a regional climate model, different combinations of variables may be necessary to adequately adjust the simulated absorbing aerosol properties of the model. In our case, $SSA_{550}$ seems to have more impact
at global scale in comparison to $AAOD_{550}$. Globally ABSORB2 has better ME and MAE than ABSORB1. The most likely explanation lies within the correlations in the observational uncertainty of the assimilated variables. In FigureS 10 the estimated POLDER errors show that the correlation between errors in $AOD_{550}$ and $AAOD_{550}$ (0.684) is higher than the correlation between errors in $AOD_{550}$ and $SSA_{550}$ (0.192), indicating that the latter combination of variables can provide better results in the data assimilation framework of this study.

**4.4 LETKF Sensitivity Experiments**

In this subsection, we investigate the sensitivity of the data assimilation system to the number of ensemble members, the localization scale, and the inflation parameter. More ensemble members (higher $n_{ens}$) provides a more accurate and detailed description of the model uncertainty at the cost of computational resources required. Previous studies with successful assimilation experiments have used $n_{ens}$ values that ranged between 12 and 80 members (Dai et al., 2019; Lin et al., 2008;
Rubin et al., 2016; Schutgens et al., 2010b; Sekiyama et al., 2010; Di Tomaso et al., 2017). In all cases, doubling $n_{ens}$ did not significantly improve the assimilation results (Schutgens et al., 2010b; Di Tomaso et al., 2017). Rubin et al. (2016) showed that by quadrupling the $n_{ens}$ (20 to 80) the RMSE was improved for most of the AERONET sites and especially for sites affected by spatially large plumes of aerosol events. Specifically, for the AERONET site Sede Boker located in southern Israel the bias and RMSE were reduced by 50% and 35% respectively. All of the core-experiments in this study consists of
$n_{ens}$=32. Two additional sensitivity experiments where conducted with $n_{ens}$=16 (SMALL) and $n_{ens}$=64 (LARGE).
Figure 19 indicates that the 64-ensemble size experiment (LARGE) managed to go a bit closer to the assimilated observations in all variables. The system managed to use the new ensemble correlations to simultaneous match a bit better the three assimilated observations. Contrary, the 16-ensemble size experiment (SMALL) difference between the assimilated observations was higher in comparison to both the 32 and the 64 ensemble size experiments for the aforementioned reason.
On the other hand, the comparison to AERONET in Figure 20 does not reveal consistent improvements for any of the variables when increasing the ensemble size. Obviously, the downside of the 64-ensemble size experiment is that it had to use double the resources and take more than twice the time to complete in comparison to the 32 ensemble size experiment, while the improvements were limited and apparent only when compared to the assimilated observations.
In the same plots the spatiotemporal collocation between the assimilated grid cell and the observations that affect it in
LETKF is tested with the $L_x$ and $L_y$ factors. Theoretically, a larger ensemble size may be able to benefit from information





coming from more distant observations (Miyoshi and Yamane, 2007; Schutgens et al., 2010b). Two additional experiments have been conducted, using similar ensemble sizes (32 and 64) as in the previous sensitivity experiments, but with higher ($L_x=6$ and $L_y=3$) than the default ($L_x=4$ and $L_y=2$). By comparing the experiments TOTAL and LOCAL1 as well as LARGE and LOCAL2 (same ensemble size, different localization factors) we can assess the effect of $L_x$ and $L_y$ factors. In both cases

the evaluation against AERONET shows that the experiment TOTAL and LARGE (core-experiments) are superior in terms of global ME and MAE, but the evaluation against POLDER shows the opposite. By comparing LOCAL1 and LOCAL2 we can conclude that higher localization factors can benefit from higher ensemble size based on the comparison with POLDER, but yet again the evaluation against AERONET has contradicting results depending on the variable.

The final experiments (INFLATE1 and INFLATE2) test the inflation parameter ($\rho$) of the LETKF, which is multiplied with

the background covariance matrix and prevents the ensemble spread of becoming too small. In each assimilation cycle, the ensemble spread of the analysis decreases, since all the ensemble members are converging to the same assimilated observations. This can create a background uncertainty that may be unrepresentative of the real background uncertainty, which will lead to an assimilation failure (Schutgens et al., 2010b). Here we test if our spatially correlated perturbations methodology, which was used to describe the model uncertainty, can keep the ensemble spread big enough for the

assimilation of POLDER observations. INFLATE1 experiment uses $\rho=1$, which basically disables the inflation feature, INFLATE2 uses $\rho=1.5$, while the rest of the experiments are using $\rho=1.1$. For a direct comparison of the inflation impact, INFLATE1 and INFLATE2 should be compared with TOTAL. Both against POLDER and AERONET, TOTAL is in most cases a bit better than INFLATE1, thus concluding that inflation ($\rho=1.1$) is a positive feature for the current assimilation framework. Additionally, when comparing INFLATE2 with TOTAL, the INFLATE2 mean error for POLDER $AOD_{550}$,

$AOD_{865}$ and $ANG_{550-865}$ is slightly smaller than in TOTAL, but in all other cases (variables, statistics, observations) TOTAL is better.

## 5. Discussion & Conclusions

We have presented the development of the first assimilation system for the global aerosol climate model ECHAM-HAM, and demonstrated successful assimilation of multiple aerosol optical properties retrievals from the POLDER SRON

algorithm using an ensemble Kalman filter. The assimilation system uses an ensemble of perturbed simulations to define model uncertainty. The ensemble is created by perturbing the emission fluxes of all aerosol species and wind field of each ensemble member with spatially correlated perturbations. The uncertainty of POLDER observations was defined by evaluating the satellite retrievals with AERONET. The forecast output based on the most recent analysis of all the experiments is compared with POLDER (not yet assimilated), AERONET and MODIS observations.

The experiment in which only POLDER $AOD_{550}$ is assimilated demonstrates considerable improvement in $AOD_{550}$, against all observations (POLDER, MODIS-DT over land, MODIS-DB, AERONET) except MODIS-DT over ocean. Furthermore, the $AOD_{550}$ correction improves also the simulated size representation through the reduction of $AE_{550-865}$ global ME by 0.13,



against POLDER observations. Contrary it is noted that in the same experiment, both the $AAOD_{550}$ and $SSA_{550}$ deteriorate in terms of global ME and MAE. In certain regions $AAOD_{550}$ was also dramatically overestimated over Africa, South America and the Atlantic Ocean. These results reveal that AOD-only assimilation may lead to large discrepancies of the simulated aerosol absorbing optical properties.

Several other experiments that assimilated a combination of $AOD_{550}$ with $AOD_{865}$, $AE_{550-865}$, $AAOD_{550}$, $SSA_{550}$ showed consistent improvement in the assimilated variables in comparison to the no-assimilation experiment. The experiment where $AOD_{550}$ & $AE_{550-865}$ & $SSA_{550}$ were assimilated simultaneously was the most promising. The difference between model fields and assimilated observations decreased for virtually all aerosol optical properties in comparison to the experiment where only $AOD_{550}$ was assimilated. The evaluation against AERONET showed that for all variables (except $SSA_{550}$) both the global ME and MAE were improved in comparison to the CONTROL experiment, demonstrating that our data assimilation system can successfully constrain the simulated aerosol burden, size and absorption properties simultaneously. Our results suggest that it is very important to consider including AE and SSA, or other properties related to aerosol size and absorption, in future operational assimilation applications and especially in reanalysis simulations. Otherwise aerosol size and absorption may be misrepresented.

Sensitivity experiments on the type and combination of the assimilated observations has been conducted. Assimilating AOD & AE instead of two AODs in different wavelengths reduces AE bias more, while assimilating AOD & SSA instead of AOD & AAOD decreases the bias of SSA and AAOD more. These results are most probably related to the correlations of the assimilated variables that the data assimilation system does not account for. Furthermore, the LETKF sensitivity experiments indicated that there was only a limited effect when varying the ensemble size, localization and inflation. Keeping the values of these parameters within the reported range of past literature, provides similar results and it is safe to say that they don't significantly affect the assimilation performance.

This work concludes that it is crucial to assimilate AE and SSA along with AOD in order to accurately correct for the burden, size and absorption of aerosol particles. The assimilation of other observational retrievals like the effective particle radii (size), column number, the fraction of spheres and the refractive index could improve the aerosol representation in the model further. Accurate measurements of these properties are expected from the SPEXone instrument on the NASA PACE Mission (Hasekamp et al., 2019a; Werdell et al., 2019). In addition, the assimilation of AE at different wavelengths could also be an interesting experiment since AE at high wavelengths (e.g. 865nm) is sensitive to the fine mode fraction and not the effective radius of aerosol particles and vice versa for AE at low wavelengths (e.g. 440nm) (Schuster et al., 2006).

**Acknowledgements**

We thank the Principal Investigators, Co-Investigators and their staff for establishing and maintaining the AERONET sites used in this investigation. The authors would like to thank Paul Snijder for providing the spatiotemporal collocated data





between POLDER ($18 \times 18$ km$^2$) and AERONET. First author is funded by a NWO/NSO project "AEROSOURCE:
Estimation of Aerosol Emissions from Polarization Data" (grant nr. ALWGO.2017.008).

**Appendix A**

POLDER retrievals with horizontal resolution $18 \times 18$ km$^2$ were spatiotemporally collocated with AERONET V3 L1.5
Aerosol Inversion dataset within an hour for the period 2006 to 2009. The AERONET L1.5 were used in order to acquire
more collocated points between POLDER and AERONET. Undoubtedly, this choice may cause an overestimation of
POLDER uncertainties for AAOD$_{550}$ and SSA$_{550}$, since AERONET L1.5 includes retrievals of aerosol absorbing optical
properties in cases of AOD$_{440}$<0.4, that are less accurate (Lacagnina et al., 2015). The AOD and AAOD of AERONET have
been converted to POLDER wavelengths (550nm, 865nm) using AE:

$$AE_{\lambda1-\lambda2} = \frac{\log\left(AOD_{\lambda1}/AOD_{\lambda2}\right)}{\log\left(\lambda1/\lambda2\right)}$$

while SSA was calculated by combining AOD and AAOD:

$$SSA = \frac{AOD - AAOD}{AOD}$$

where $\lambda1$ and $\lambda2$ the wavelengths in nm. Afterwards the relative errors (POLDER–AERONET / POLDER) for AOD$_{550}$,
AOD$_{865}$ and AAOD$_{550}$ were plotted against POLDER AOD$_{550}$, AOD$_{865}$ and AAOD$_{550}$ respectively (FigureA 1). In a similar
fashion the errors of AE$_{550-865}$ and SSA$_{550}$ (POLDER–AERONET) where plotted against POLDER AOD$_{550}$. In each case the
XX' axis was partitioned in six bins and the standard deviation of the errors was calculated for each bin. Lastly, the standard
deviation of the relative differences for each bin and the variables AOD$_{550}$, AOD$_{865}$ and AAOD$_{550}$ was multiplied with the
POLDER $1° \times 1°$ gridded dataset to represent the POLDER uncertainty. While in the case of AE$_{550-865}$ and SSA$_{550}$ the
standard deviation of the absolute errors for each bin was set as POLDER uncertainty. This fairly simple representation of
POLDER uncertainties may be imprecise for some specific areas (e.g. the high albedo arid areas) or some remote areas (e.g.
over ocean), but carries the advantage of being based on high quality independent observations.

**Appendix B**

$$ME = \frac{1}{N}\sum_i sim_i - obs_i$$

$$MAE = \frac{1}{N}\sum_i |sim_i - obs_i|$$

$$RMSE = \sqrt{(sim_i - obs_i)^2}$$

where sim and obs are the simulated and observed values while N is the population.




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





**Figure 1.** The uncertainty and the relative uncertainty of POLDER for AOD$_{550}$ (a,b) AOD$_{865}$ (c,d), AAOD$_{550}$ (e,f) and the uncertainty for AE$_{550-865}$ (g) and SSA$_{550}$ (h) averaged over the period 20$^{th}$ of July to 28$^{th}$ of August 2006. The global mean of each variable is denoted in the bottom-right corner for each case.





**Table 1. State vector of the assimilation system composed by the mass mixing ratio of all the modes and species and number mixing ratio (PN) for all modes in ECHAM-HAM. Aerosol in nucleation mode are not considered in the assimilation since they contribute a very small fraction in the optical depth of aerosols.**


| | | SO$_4$ | BC | OC | SS | DU | PN |
|---|---|---|---|---|---|---|---|
| **Soluble** | Nucleation | ✕ | | | | | ✕ |
| | Aitken | ✓ | ✓ | ✓ | | | ✓ |
| | Accumulation | ✓ | ✓ | ✓ | ✓ | ✓ | ✓ |
| | Coarse | ✓ | ✓ | ✓ | ✓ | ✓ | ✓ |
| **Insoluble** | Nucleation | | | | | | |
| | Aitken | | ✓ | ✓ | | | ✓ |
| | Accumulation | | | | | ✓ | ✓ |
| | Coarse | | | | | ✓ | ✓ |

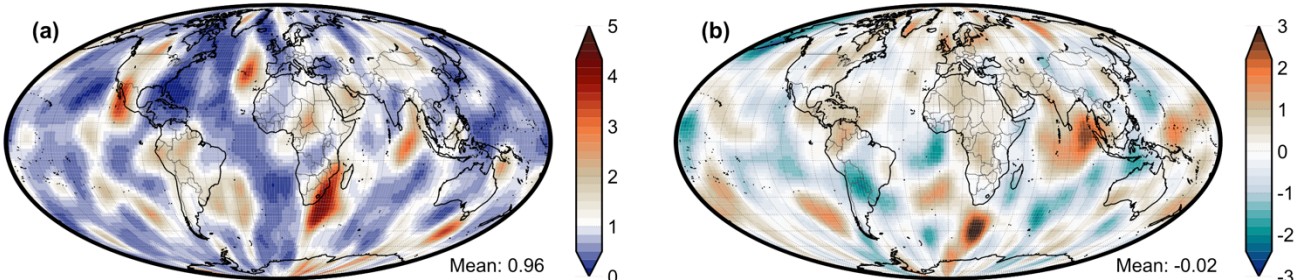

**Figure 2. An example of the spatially correlated perturbations maps generated for an ensemble member regarding (a) dust emissions and (b) U-component of the wind.**



**Figure 3. The uncertainty and the relative uncertainty of ECHAM-HAM for AOD$_{550}$ (a,b) AOD$_{865}$ (c,d), AAOD$_{550}$ (e,f) and the uncertainty for AE$_{550-865}$ (g) and SSA$_{550}$ (h) averaged over the period 20$^{th}$ of July to 28$^{th}$ of August 2006. The uncertainty is estimated using the standard deviation of an ensemble (32 members) where each member used different spatial correlated perturbations on aerosol emissions and wind. The relative uncertainty is defined as the ratio of standard deviation to mean. The global mean of each variable is denoted in the bottom-right corner for each case.**



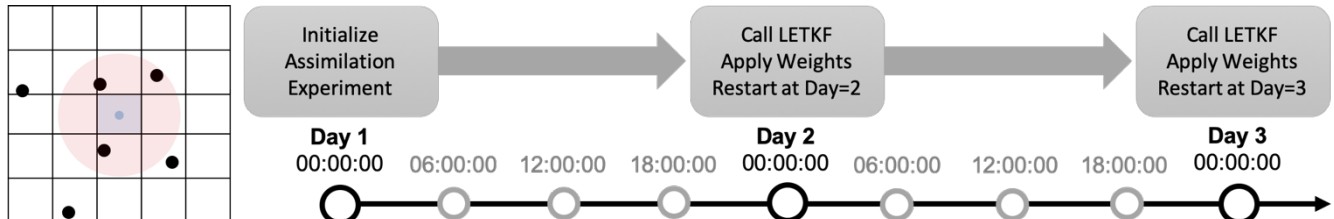

**Figure 4. On the right side the grid cell scale collocation of the assimilated grid and the surrounded observations is illustrated. The blue highlighted area denotes the assimilated grid cell, the red circle around it represents the npatch distance and black dots represent the observations. It is noted that in LETKF the npatch distance is represented in grid cell units, but for illustrative purposes here is represented as a circle. On the left side the daily assimilation cycle is depicted for 3 days.**


**Table 2. The name along with the assimilated aerosol optical properties and the physical meaning of each experiment. All the assimilated parameters are retrievals of POLDER. All of the experiments used the rescaled emission factors of subsection 3.4.**

| | Name | Assimilated Parameters | Physical Meaning |
|---|---|---|---|
| **Core-Experiments** | CONTROL | - | An ensemble of perturbed simulations consisted of 32 members that represent the background error. |
| | MASS | $AOD_{550}$ | Correction of total aerosol mixing ratio. |
| | SIZE1 | $AOD_{550}$ & $AOD_{865}$ | Correction of total aerosol mixing ratio and aerosol size distribution. |
| | SIZE2 | $AOD_{550}$ & $AE_{550\text{-}865}$ | Correction of total aerosol mixing ratio and aerosol size distribution. |
| | ABSORB1 | $AOD_{550}$ & $AAOD_{550}$ | Correction of total aerosol mixing ratio and absorbing aerosol mixing ratio. |
| | ABSORB2 | $AOD_{550}$ & $SSA_{550}$ | Correction of total aerosol mixing ratio and absorbing aerosol mixing ratio. |
| | TOTAL | $AOD_{550}$ & $AE_{550\text{-}865}$ & $SSA_{550}$ | Correction of total aerosol mixing ratio, aerosol size distribution and absorbing aerosol. |
| **Sensitivity Experiments** | SMALL | $AOD_{550}$ & $AE_{550\text{-}865}$ & $SSA_{550}$ | Ensemble size 16. |
| | LARGE | $AOD_{550}$ & $AE_{550\text{-}865}$ & $SSA_{550}$ | Ensemble size 64. |
| | LOCAL1 | $AOD_{550}$ & $AE_{550\text{-}865}$ & $SSA_{550}$ | Ensemble size 32, Lx=6 and Ly=3 |
| | LOCAL2 | $AOD_{550}$ & $AE_{550\text{-}865}$ & $SSA_{550}$ | Ensemble size 64, Lx=6 and Ly=3 |
| | INFLATE1 | $AOD_{550}$ & $AE_{550\text{-}865}$ & $SSA_{550}$ | Ensemble size 32 and Inflation=1 |
| | INFLATE2 | $AOD_{550}$ & $AE_{550\text{-}865}$ & $SSA_{550}$ | Ensemble size 32 and Inflation=1.5 |



**Figure 5. Aerosol Optical Depth at 550nm for (a) POLDER, the experiments (b) CONTROL, (c) MASS, (d) SIZE2, (e) ABSORB2, (f) TOTAL and their differences (Model-POLDER, g-l). All fields are spatiotemporally collocated to the available measurements of POLDER for the period 20th of July to 28th of August 2006. Grey-filled grid cells indicate the absence of any valid POLDER measurements for the study period.**

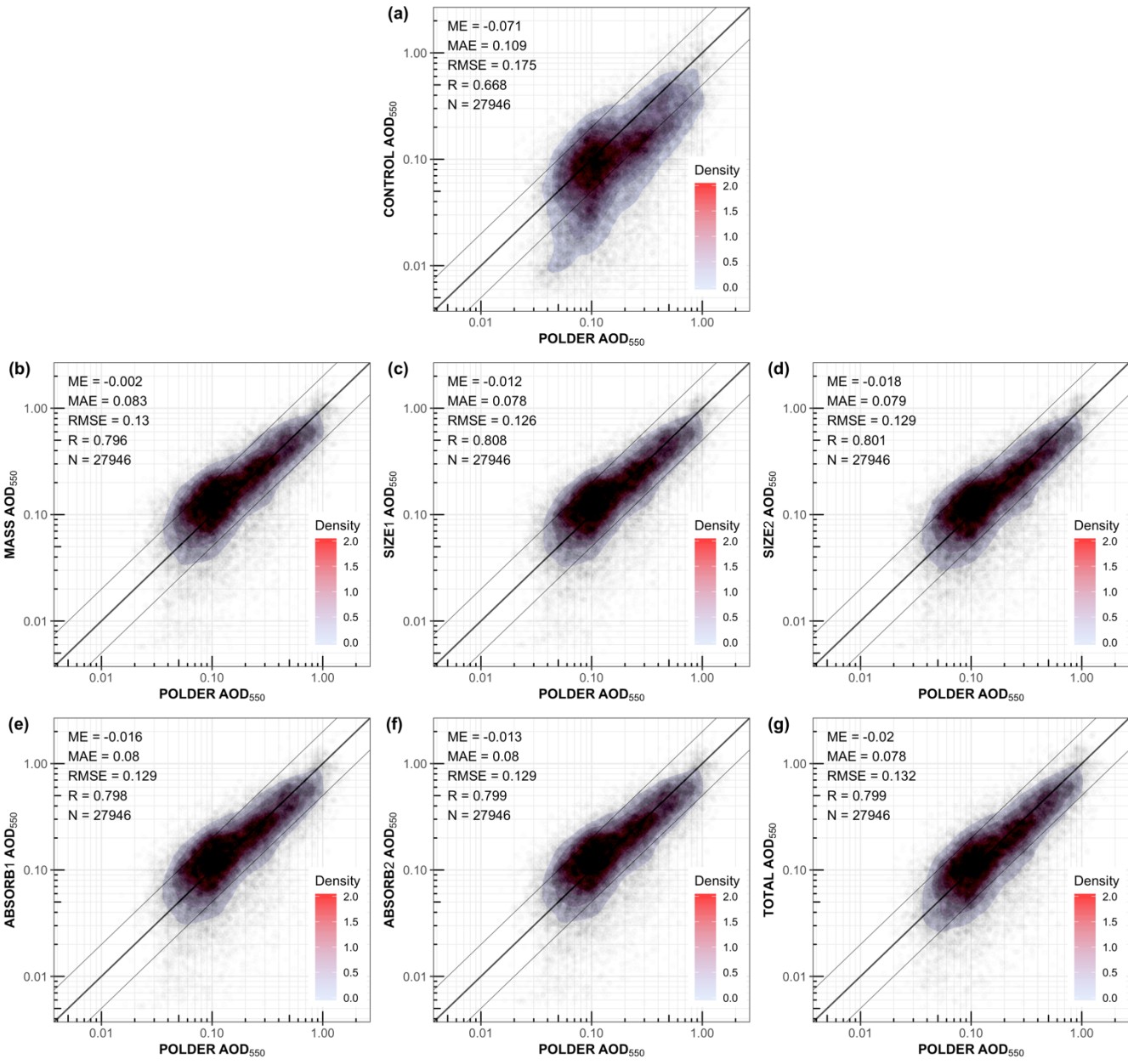

**Figure 6. Aerosol Optical Depth at 550nm of POLDER against the core-experiments, regarding all the available points of POLDER for the period 20th of July to 28th of August 2006. In each subplot the bold line indicates the perfect model (y=x), while the two thinner lines confine the -200% and 200% bias boundaries respectively. The shade depicts the density of points. N represents the number of total points.**





**Figure 7.** Similar to Figure 5 but for Angstrom Exponent 550-865nm.







**Figure 8. Similar to Figure 6 but for Angstrom Exponent 550-865nm. The two thinner lines confine the -30% and 30% bias boundaries respectively.**





090    **Figure 9. Similar to Figure 5 but for Absorption Optical Depth at 550nm.**



**Figure 10. Similar to Figure 5 but for Single Scattering Albedo at 550nm.**





**Figure 11. Similar to Figure 6 but for Aerosol Absorption Optical Depth at 550nm.**





1095

**Figure 12. Similar to Figure 6 but for Single Scattering Albedo at 550nm. The two thinner lines show the -10% and 10% bias boundaries respectively.**




**Table 3. Global column burden (Tg) of all aerosol species regarding the experiment CONTROL and the induced percentage changes due to assimilation for the MASS, SIZE2, ABSORB1 and TOTAL experiment.**

| Species | CONTROL | MASS | SIZE2 | ABSORB1 | ABSORB2 | TOTAL |
|---------|---------|------|-------|---------|---------|-------|
| DU | 11.88 | +34% | +25% | +26% | +35% | +49% |
| SS | 2.21 | 0% | +3% | 0% | -3% | +20% |
| OC | 2.04 | +112% | +89% | +78% | +95% | +65% |
| BC | 0.17 | +396% | +256% | +27% | +102% | +95% |
| SO$_4$ | 2.11 | +101% | +67% | +112% | +121% | +30% |

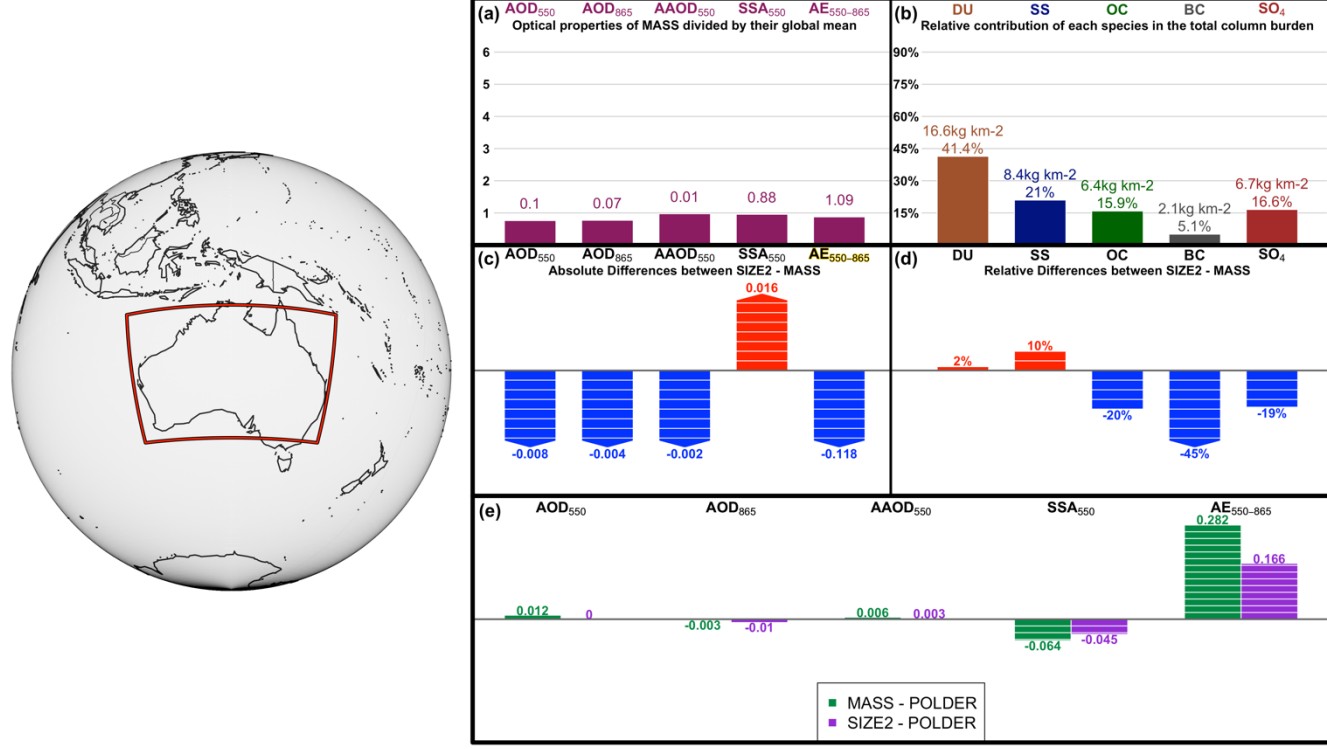

**Figure 13. An overview of the changes due to the addition of POLDER AE$_{550-865}$ in the assimilation for Australia. (a) Depicts the mean values of five aerosol optical properties while the height of bars indicate the regions relative amount of each property in comparison to the global mean. (b) Illustrates the column burden of five aerosol species as simulated in the experiment MASS. The percentage in the column burdens of each species specifies the relative contribution of each species in the total column burden. (c) Illustrates the absolute changes caused in aerosol optical properties, due to the assimilation of AE$_{550-865}$ (SIZE2 − MASS), while the height of bars indicate if the change is just positive or negative. (d) Shows the relative changes caused in the mixing ratio for each species, due to the assimilation of AE$_{550-865}$ (SIZE2 − MASS). (e) Displays the aerosol optical properties bias in comparison to POLDER for the experiments MASS and SIZE2.**



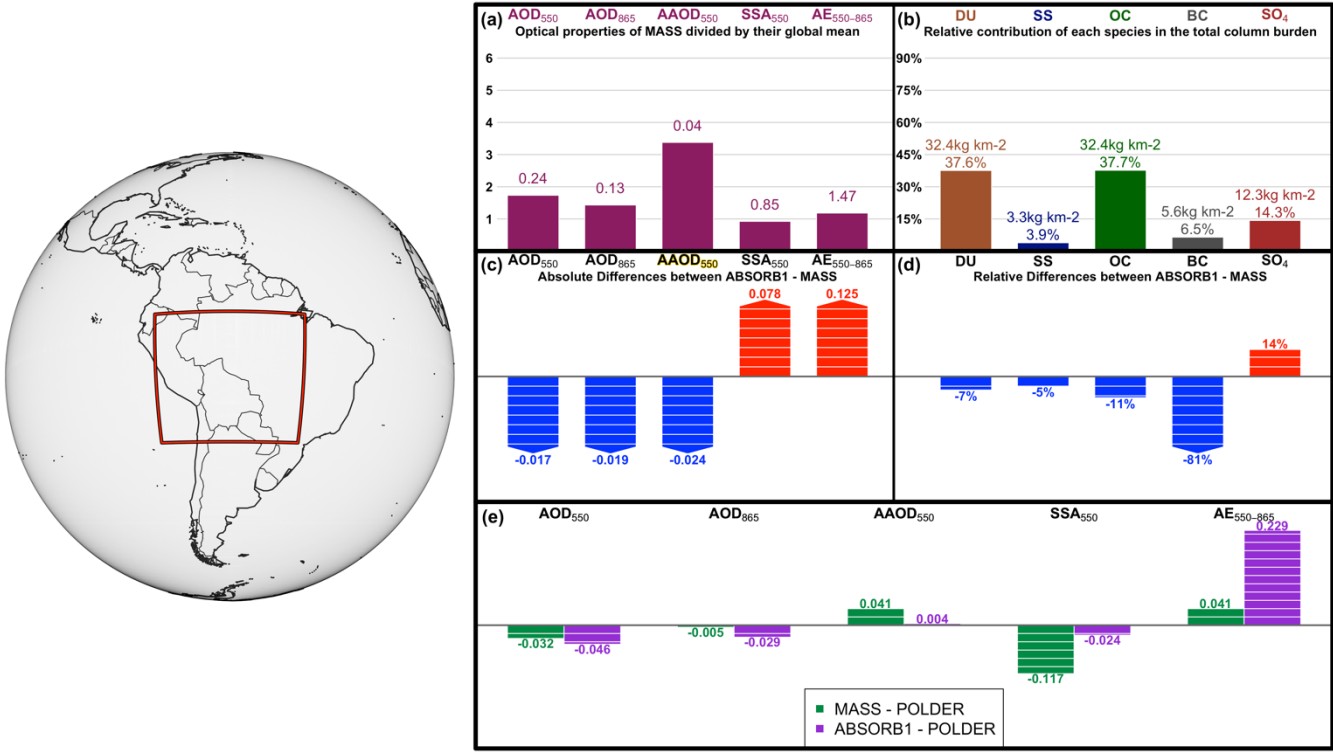

**Figure 14. Similar to Figure 13 but for South America. The depicted experiments are MASS and ABSORB1, hence the changes in subplot c and d are based on the addition of AAOD$_{550}$ to the assimilation.**


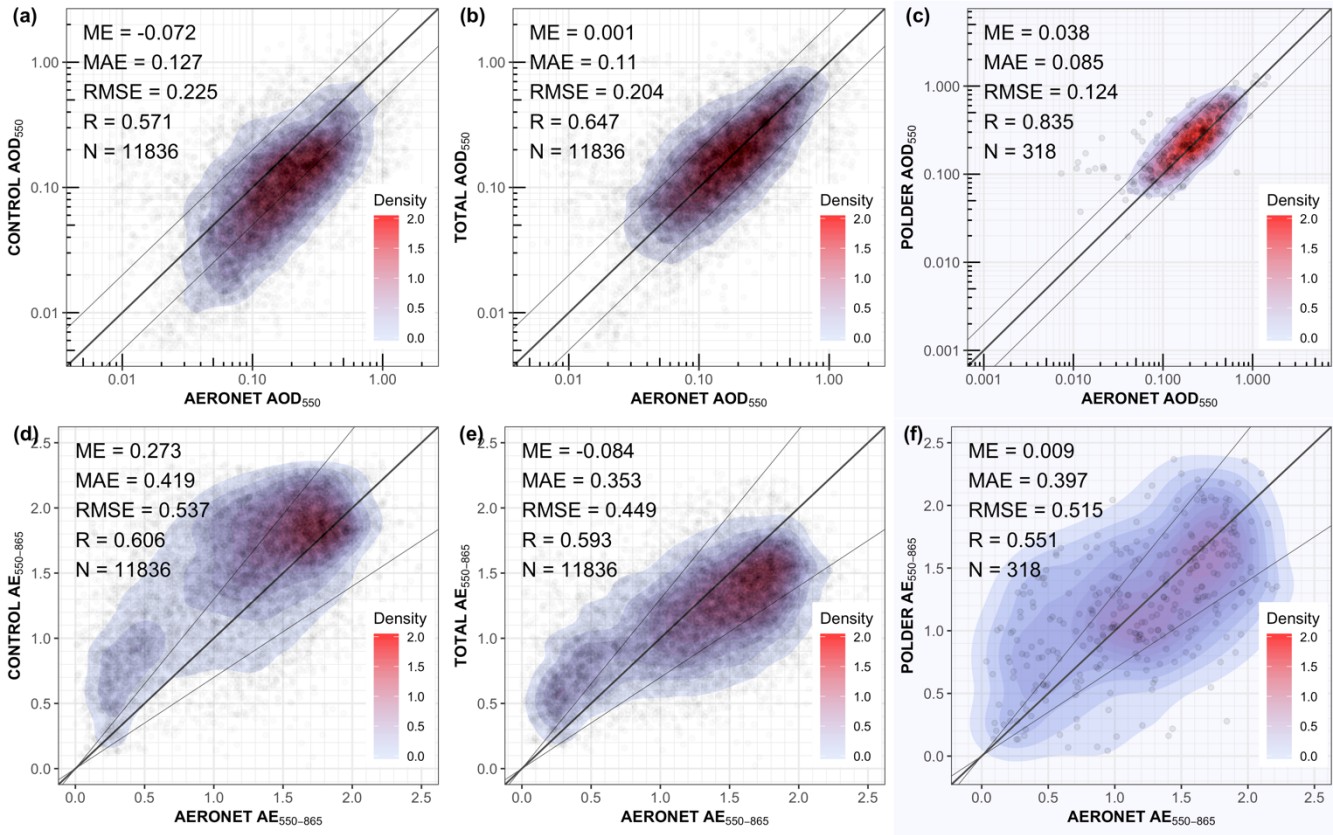

**Figure 15.** Comparison between the AERONET (Direct-Sun V3 L2) and the experiments CONTROL, TOTAL and POLDER for AOD$_{550}$ (a, b, c) and AE$_{550\text{-}865}$ (d, e, f).





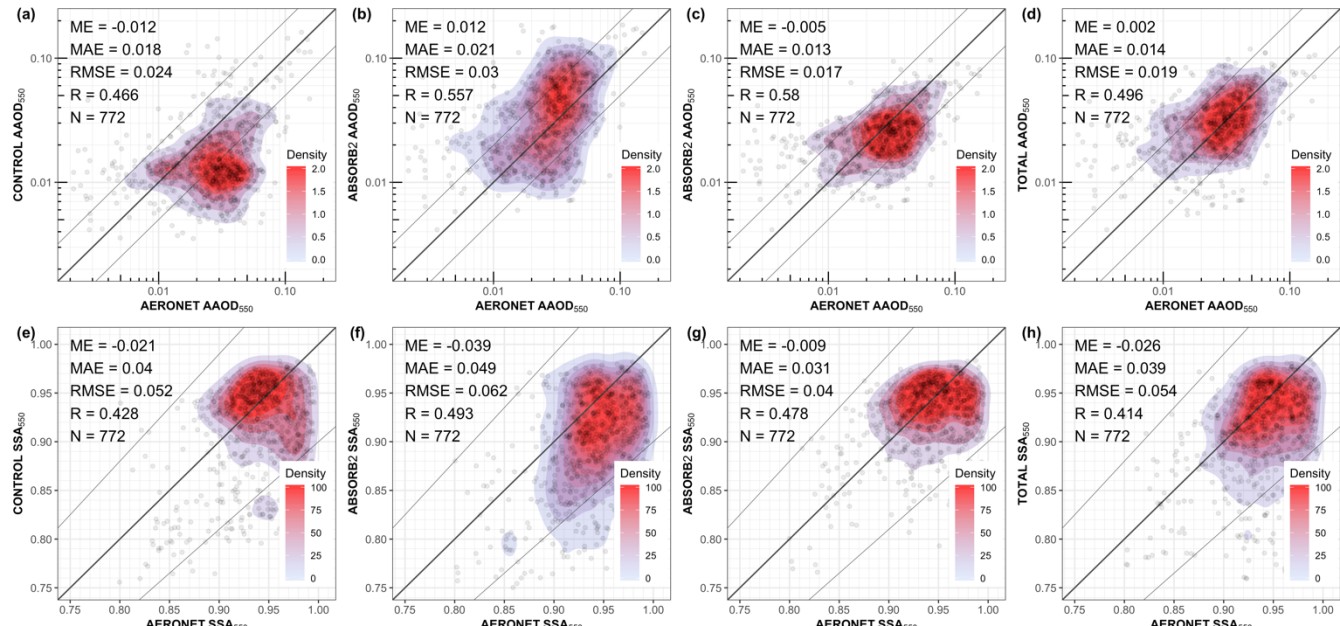

**Figure 16.** Comparison between the AERONET (Aerosol Inversion V3 L2) and the experiments CONTROL, MASS, ABSORB2 and TOTAL for $AAOD_{550}$ (a, b, c, d) and $SSA_{550}$ (e, f, g, h).





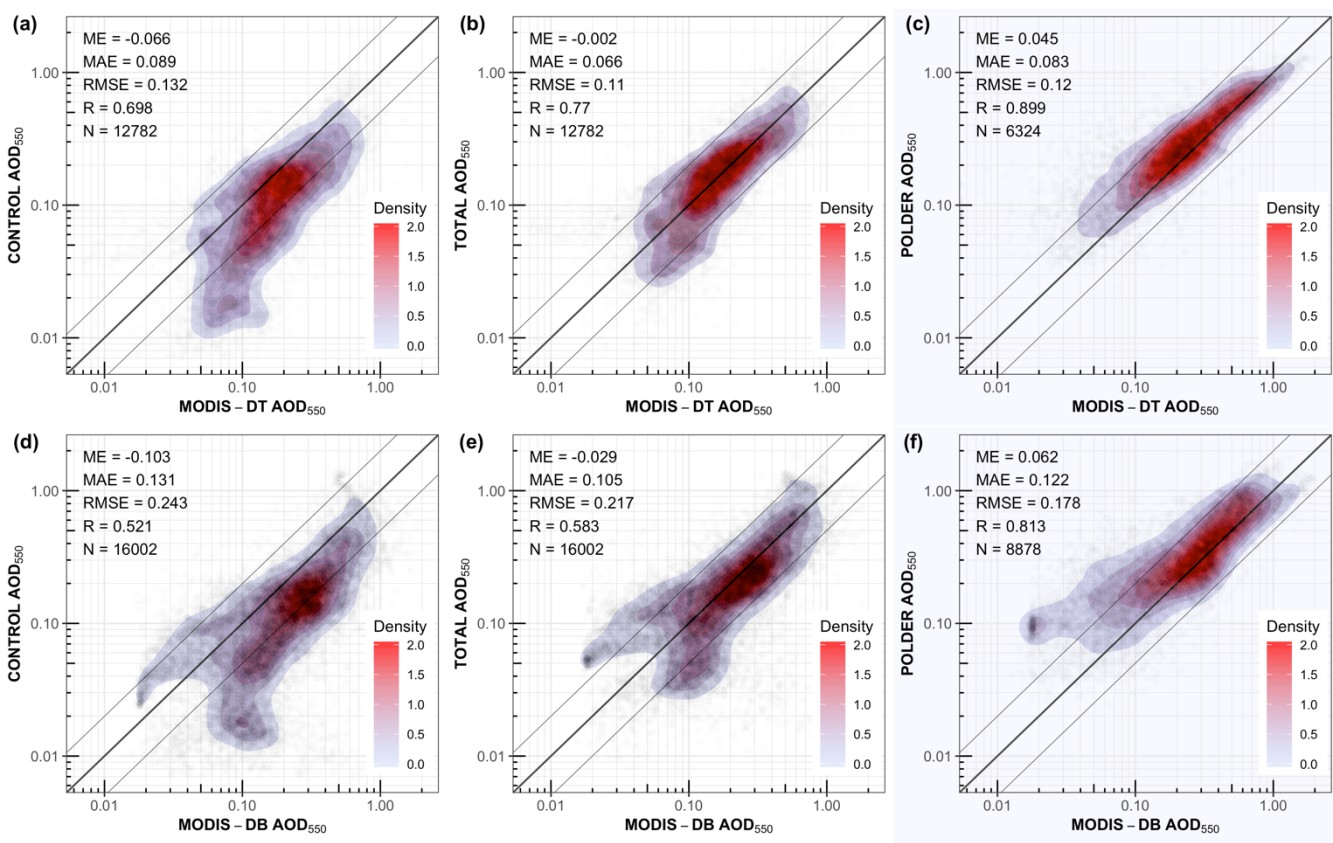

120

**Figure 17. The AOD$_{550}$ of MODIS-DT over land against the experiments CONTROL, TOTAL and POLDER (a,b,c). Similar for MODIS-DB AOD$_{550}$ (d,e,f).**

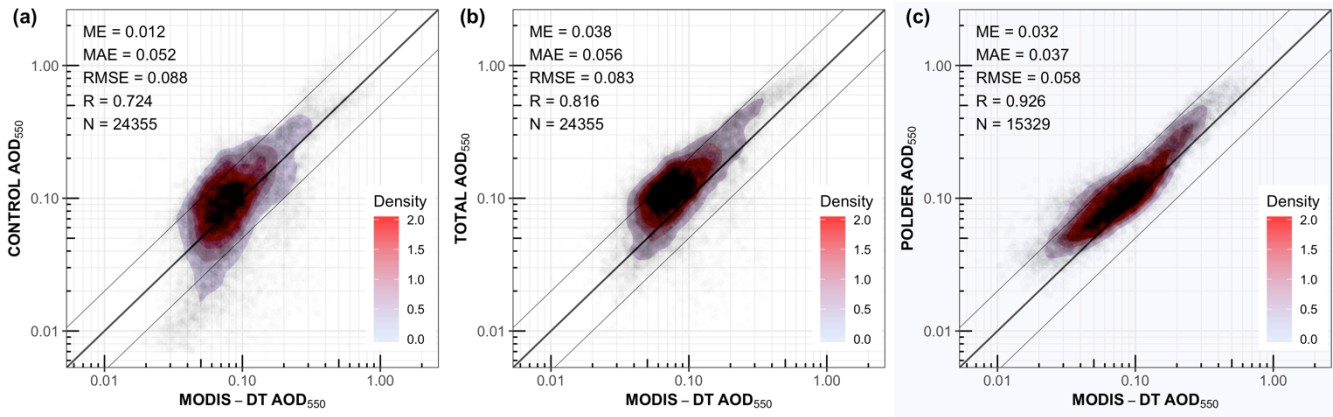

**Figure 18. The AOD$_{550}$ of MODIS-DT over ocean against the experiments CONTROL, TOTAL and POLDER (a,b,c).**


125

**Figure 19. ME and MAE for all the spatiotemporal collocated points between POLDER (assimilated observations) and the experiments, averaged over for the whole study period.**



**Figure 20. ME and MAE for all the spatiotemporal collocated points between AERONET (independent observations) and the experiments. AOD$_{550}$, AOD$_{865}$ and AE$_{550-865}$ are calculated using the AERONET – Direct-Sun V3 L2 while AAOD$_{550}$ and SSA$_{550}$ using the AERONET Aerosol Inversion V3 L2.**

1130







**FigureA 1. POLDER AOD$_{550}$, AOD$_{865}$, and AAOD$_{550}$ uncertainty estimation using the relative errors of POLDER-AERONET and POLDER AE$_{550-865}$ and SSA$_{550}$ uncertainty estimation using the differences of POLDER-AERONET. Each variable is partitioned in six bins along the XX' axis. The grey shade indicates the Standard Deviation (SD) while the purple shade illustrates the distribution for each bin.**

135