# Peer review of "Assimilating aerosol optical properties related to size and absorption from POLDER/PARASOL with an ensemble data assimilation system"

_Atmospheric Chemistry and Physics, 2020_

## Referee Comment (RC1) · Anonymous Referee #2 · 1 Nov 2020

This study presents an ensemble Kalman filter-based data assimilation system developed for the ECHAM-HAM and applied to POLDER derived observations of optical properties. This paper assesses the added value of assimilating AE, AAOD and SSA, in addition to AOD. The experiment where POLDER AOD, AE and SSA are assimilated shows systematic improvement in mean error, mean absolute error and correlation for AOD, AE, AAOD and SSA compared to the experiment where only AOD is assimilated. The paper is well written and easy to follow. I only have some minor concerns listed below and recommend the paper for publication after these concerns are addressed.

[Figure]

Figure 1a: Why are there no AOD retrievals available over India and why are the uncertainties larger in the Southern Ocean?

Line 49: Change "disentangles" to "disentangle".

Line 67: Change "colour, polarization" to "colour, and polarization"

Lines 110-112: Can you provide an estimate of how many data points you gain by using the L1.5 AERONET retrievals rather than L2 retrievals. What is the effect of using L1.5 AOD retrievals on the POLDER uncertainty estimates?

Line 161: Change "∼0.03 is" to "∼0.03 in".

Line 184: Since the model resolution is (1.875° x 1.875°) and the POLDER resolution is 1 x 1 degrees, do you use some kind of super observation approach for the assimilation?

Line 343-344: Since you assume the same level of uncertainty for both the natural and anthropogenic aerosols, does this approach not underestimate the background error covariance?

Line 353: Change "initially" to "initial".

Line 382: What is Section 0?

Line 386: I am not sure if I understood the daily assimilation set-up correctly. Do you run the daily forecast from 00 to 23 hour first and then call the LETKF code for the assimilation of POLDER observations at 00, 06, 12, and 18 hours? Does the next day forecast use initial conditions from the 18 h assimilation? If this is correct, what is the benefit of assimilation at 00, 06, ad 12 hours because we are not accumulating the benefits of assimilation at these times in the forecasts.

Figure 15: The correlation coefficient for AE decreases from the Control to Total experiment. What is the reason for that? Is it because POLDER AE has a lower correlation coefficient compared to control?

Figure 16: Should the title of y-axis be MASS AAOD in panels b and f?

[Figure]

---

## Referee Comment (RC2) · Anonymous Referee #1 · 5 Dec 2020

Review of "Assimilating aerosol optical properties related to size and absorption from POLDER/PARASOL with an ensemble data assimilation system" by Tsikerdekis et al.

The authors tried to implement Kalman Filter technique into a global aerosol model (ECHAM-HAM) and performed various assimilation experiment with aerosol optical properties derived by POLDER to investigate the impact of assimilating multiple AODs, AE, AAOD, and SSA in addition to AOT in single wavelength. They found that the additional information achieved further improvement in aerosol forecasting compared to the forecast where only AOD is assimilated. I found that this paper is well written, will

be of interest to the scientific community and suitable for publication in ACP with minor revision.

Specific comments:

L121 and L203 Both the retrieval algorithm of POLDER product and the calculation processes of aerosol optical properties in the model include many assumptions (e.g., aerosol model, size distribution, and refractive index etc.). These basic assumptions are consistent? If not, how did the differences affect the assimilation results.

L345 The authors used randomly perturbed wind to make ensemble members. How about air mass? The wind perturbed method can keep conservation of mass and mechanical equilibrium (e.g., geostrophic balance) produced in ERA-interim?

L421 It is well known that dust emissions have large inter-annual (seasonal) variations. My concern is that the yearly-mean based rescaling generate additional biases in the simulation.

L459 Subsection 4.2?

L477 Figure 5i should be Figure 5l?

L477 At first, could you explain why MASS caused a large positive bias in the South Atlantic that CONTROL did not cause.

L484 You did not show any validation result about aerosol mass mixing ratio.

L516-518 What does this mean? Was there problem in BC simulation (e.g., refractive index)? Did model underestimate other aerosol species (e.g., organic aerosols)? Could you make this clear?

Figure 4 It's the wrong way around.

Figure 17f Why does MODIS-DB underestimate AODs where POLDER estimates AOD as about 0.1?
[Figure]

---

## Author Comment (AC1) · 18 Dec 2020

Response to Referee #2 for the manuscript: "Assimilating aerosol optical properties related to size and absorption from POLDER/PARASOL with an ensemble data assimilation system"

Dear Editor & Reviewers,

Thank you for reviewing our submitted manuscript. Your comments helped us highlight and clarify some of our results better. Below you can find our responses for all of the raised questions.

Best regards,
Athanasios Tsikerdekis

**Format:**
Question
Answer
Quoted text, **added/changed text**,

**Minor revisions:**

Figure 1a: Why are there no AOD retrievals available over India?

The uncertainties shown in Figure1 are averaged over the period 20th of July to 28th of August 2006, which is during the summer monsoon in India where high precipitation and cloudiness occur. Therefore, the lack of retrievals over India is caused by the cloud screening of the algorithm. Note that multi-angle multi-wavelength photopolarimetric measurements have the ability to distinguish scattering caused by aerosol particles and cloud droplets (Stap et al., 2015) which facilitates cloud screening.

Figure 1a: Why are the uncertainties larger in the Southern Ocean?

The uncertainty of POLDER observations is defined after evaluating it with AERONET (FigureA1). Low AOD values (<0.05) have a very high relative uncertainty (~100%), but small absolute uncertainty. Southern ocean has very low AOD (<0.05) values over the whole course of the evaluated period. As noted in the manuscript, AERONET is a spatially sparse ground-based network of stations some generalization had to be made as far as the performance of POLDER in remote areas. The southern ocean is definitely one of these remote areas.

Lines 110-112: Can you provide an estimate of how many data points you gain by using the L1.5 AERONET retrievals rather than L2 retrievals. What is the effect of using L1.5 AOD retrievals on the POLDER uncertainty estimates?

POLDER uncertainty estimates were calculated using the AERONET Inversion L1.5 V3 dataset in cases were all variables were available or could be calculated ($AOD_{550}$, $AOD_{865}$, $AE_{550-865}$, $AAOD_{550}$, $SSA_{550}$). There are two major differences between L2.0 and L1.5. (i) Improved cloud screening in L2.0 and (ii) AAOD and SSA calculation use only cases of AOD>0.4 in L2.0. If L2.0 data were used instead of L1.5 then we would have ~4 times less available collocated data points between POLDER and AERONET. Our POLDER uncertainty estimates are conservative because all POLDER – AERONET differences are attributed to POLDER while in some cases half of it comes from AERONET (for other properties than AOD, for AOD itself the AERONET uncertainty is 0.01).

Line 184: Since the model resolution is (1.875° x 1.875°) and the POLDER resolution is 1 x 1 degrees, do you use some kind of super observation approach for the assimilation?

The current version of the data assimilation system does not account for representations errors. We are planning to include a method that account for that. However it is noted that cloudiness is the biggest driver in representation errors in AOD. Thus if the clouds are represented correctly in the model the representation error would be zero. It is noted that ECHAM-HAM calculates aerosol optical properties on cloud free part of each grid and POLDER retrieves only in non-cloudy conditions.

Line 343-344: Since you assume the same level of uncertainty for both the natural and anthropogenic aerosols, does this approach not underestimate the background error covariance?

Thank you for this comment. Ideally we should have picked different uncertainty for natural and anthropogenic sources, though the version of the data assimilation used in the paper could perturbed emissions only by species and not by emission sector (e.g. Industrial, ships, fires etc), thus the distinction of anthropogenic and natural sources for some species (e.g. OC, BC, SO4, SO2) was not possible at the moment. In our follow-up research we are planning to perturb with high ensemble standard deviation at least the purely natural species (DU and SS).

Line 382: What is Section 0?
Corrected to subsection 3.2.

Line 386: I am not sure if I understood the daily assimilation set-up correctly. Do you run the daily forecast from 00 to 23 hour first and then call the LETKF code for the assimilation of POLDER observations at 00, 06, 12, and 18 hours? Does the next day forecast use initial conditions from the 18 h assimilation? If this is correct, what is the benefit of assimilation at 00, 06, ad 12 hours because we are not accumulating the benefits of assimilation at these times in the forecasts.

Thank you for giving me the opportunity to clarify and adjust the text. Background forecast is run from Day1 00:00:00 to Day2 00:00:00. POLDER observations are assimilated for 00, 06, 12, 18 and they adjust the mixing ratio at Day2 00:00:00 (analysis). Then the system is restarted using as initial condition the analysis conditions at Day2 00:00:00. The manuscript text was adjusted accordingly:

The daily cycle of data assimilation involves daily forecasts **(Day$_t$ 00 UTC to Day$_{t+1}$ 00 UTC)** of all perturbed ensemble members. Upon completion of these simulations, the LETKF code is called which performs a spatial collocation of the simulated (ECHAM-HAM) and the retrieved (POLDER) observations for four temporal time-steps (00, 06, 12, 18 UTC). Subsequently LETKF computes a new analysis state vector (ECHAM-HAM aerosol mixing ratio) at  **Day$_{t+1}$ 00 UTC**, which serves as initial conditions for the next day's forecast. The process is repeated till the end of the data assimilation experiment.

Figure 15: The correlation coefficient for AE decreases from the Control to Total experiment. What is the reason for that? Is it because POLDER AE has a lower correlation coefficient compared to control?

Thank you for this comment. This may indeed be the reason. However, we consider a difference between 0.593 and 0.606 not significant.

Figure 16: Should the title of y-axis be MASS AAOD in panels b and f?

Indeed that is true, thank you for pointing this out. Figure was updated.

**Changes:**

Line 49: Change "disentangles" to "disentangle".

Line 67: Change "colour, polarization" to "colour, and polarization"

Line 161: Change "~0.03 is" to "~0.03 in".

Line 353: Change "initially" to "initial".

All of the abovementioned changes have been addressed.

---

## Author Comment (AC2) · 18 Dec 2020

Response to Referee #1 for the manuscript: "Assimilating aerosol optical properties related to size and absorption from POLDER/PARASOL with an ensemble data assimilation system"

Dear Editor & Reviewers,

Thank you for reviewing our submitted manuscript. Your comments helped us highlight and clarify some of our results better. Below you can find our responses for all of the raised questions.

Best regards,
Athanasios Tsikerdekis

**Format:**

Question
Answer
Quoted text, **added/changed text**,

**Minor revisions:**

L121 and L203 Both the retrieval algorithm of POLDER product and the calculation processes of aerosol optical properties in the model include many assumptions (e.g., aerosol model, size distribution, and refractive index etc.). These basic assumptions are consistent? If not, how did the differences affect the assimilation results.

There are some differences. POLDER retrieval algorithm assumes a bimodal lognormal size distribution (fine and coarse) while ECHAM assumes seven size modes (4 for insoluble and 3 for soluble particles) of internally mixed aerosols species. However, the aerosol optical properties we assimilate (AOD, AAOD, SSA, AE) are uniquely defined so there is no ambiguity here. Furthermore in the context of SPEXone (a future Multi-Angle Polarimeter instrument like POLDER), our colleagues retrieved these aerosol optical properties using as input ECHAM-HAM fields (e.g. mixing ratio, number density, refractive index). The differences of retrieved (SPEXone algorithm) and the simulated (ECHAM-HAM) aerosol optical properties were minor. As far as the second part of this question, although very interesting, it is hard to answer within the context of this paper. It would require assimilation experiment under the framework of Observing System Simulation Experiments (OSSEs) where the assimilated properties were retrieved using different assumption in the retrieval algorithm in each experiment.

L345 The authors used randomly perturbed wind to make ensemble members. How about air mass? The wind perturbed method can keep conservation of mass and mechanical equilibrium (e.g., geostrophic balance) produced in ERA-interim?

The wind is perturbed on the nudging data of ERA-interim not on the final wind fields of ECHAM-HAM. Specifically, the wind perturbations is performed by nudging the members of the ensemble into distinct perturbed versions of ERA-interim, thus the mass and mechanical equilibrium for each member is handled by ECHAM-HAM (otherwise the model would crash). Now, the winds in the perturbed ERA-interim versions of course are not consistent in term of geostrophic balance (e.g. ERA-interim perturbed wind and not perturbed pressure do not match), since the wind perturbation was performed as a post-processing analysis on the output of ERA-interim and not by actually creating perturbed runs of ERA-interim. Ideally, the

ERA-interim wind perturbations could be created by having different version of ERA-interim where different observations have been assimilated each time, though this is not available. In our follow-on study we using ERA-5 ten analysis members to estimate wind uncertainty.

L421 It is well known that dust emissions have large inter-annual (seasonal) variations. My concern is that the yearly-mean based rescaling generate additional biases in the simulation. Thank you for that comment. This is indeed true, especially for dust emissions that fluctuate a lot seasonally. Although note that rescaling factors are only used to remove yearly biases. The timing of dust storms depend on meteorology (e.g. wind speed, soil moisture etc) and it is not affected by these rescaling factors. Plots below show the differences between POLDER – ECHAM for July (left), August (middle) and the whole year of 2006 (right). The positive emissions factor for dust over desert (~1.3) based on the differences of POLDER – ECHAM for the whole year, indeed add biases at some locations in the simulation (Western Sahara). Although in the majority of the Sahara sources, rescaling factors decrease biases, so we consider this scaling still as an improvement compared to not-scaling dust emissions. In addition we chose to use rescaling factors based on a yearly evaluation since our next goal is to expand our experiment for the year 2006.

[Figure]

L459 Subsection 4.2?
Yes, thank you for pointing this out.

L477 Figure 5i should be Figure 5l?
Yes, we have corrected it.

L477 At first, could you explain why MASS caused a large positive bias in the South Atlantic that CONTROL did not cause.
Thank you for that comment. The positive bias over South Atlantic in the MASS experiment is caused by transport. The assimilation of AOD over the Tropics increases aerosol mixing ratio which is then transported by the westward flow over South Atlantic. Now, the assimilation of AOD over South Atlantic should compensate some of that effect by decreasing aerosol mixing ratio, but the results show that assimilating just AOD is not enough to sufficiently resolve that. The inclusion of other observations in the assimilation proves that they provide the additional constrain needed to limit the overestimation of AOD over South Atlantic due to transport. The following explanation was added to the manuscript.

**Overall local biases are decreasing after assimilation, however over certain areas biases can be increased, for example over South Atlantic ocean in the MASS experiment (**Error! Reference source not found.**h). The assimilated AOD$_{550}$ over Africa in the Tropics increases the aerosol mixing ratio over land, which is then transported westward over South Atlantic. The assimilation of AOD$_{550}$ over South Atlantic should be compensating some of that effect by decreasing the aerosol mixing ratio, but evidently not sufficiently.**  The assimilation of other aerosol optical properties like AE$_{550}$-

$_{865}$ and SSA$_{550}$ reduce the South Atlantic positive AOD$_{550}$ bias, especially in the case of the TOTAL experiment (**Error! Reference source not found.**l), indicating that the simultaneous assimilation of multiple variable can **improve** the simulated AOD$_{550}$  spatial representation **in some places**.

L484 You did not show any validation result about aerosol mass mixing ratio.
Aerosol mass mixing ratio is not evaluated. But it is the state vector in the data assimilation and its adjustment leads to a better agreement in AOD. The sentence in question was modified for clarity:

The consistent improvement of AOD$_{550}$ in the assimilation experiments demonstrates  **that the inclusion of** different combination of assimilated observations **does not negatively affect AOD$_{550}$.**

L516-518 What does this mean? Was there problem in BC simulation (e.g., refractive index)? Did model underestimate other aerosol species (e.g., organic aerosols)? Could you make this clear?
Thank you for the opportunity to clarify that. The assimilation of AOD$_{550}$ will adjust the aerosol mixing ratio only based on aerosol's extinction and not on aerosol's absorption. A good example to illustrate this is the Amazon basin where CONTROL experiment underestimates AOD and overestimates AAOD. The assimilation of only AOD will lead to an increase of aerosol mixing ratio, hence AAOD will be increased too (increasing also the overestimation of AAOD further). The paragraph in the manuscript was rewritten to cause less confusion

**It is interesting to note that in the MASS experiment the AOD$_{550}$ is improved (**Error! Reference source not found.**h), but SSA$_{550}$ and AAOD$_{550}$ not so much, especially in regions like South America, Africa and the Atlantic Ocean (**Error! Reference source not found.**h, Error! Reference source not found.**h). The reason behind that is easiest to explain over South America, where AOD$_{550}$ is underestimated and AAOD (SSA) is overestimated (underestimated) in CONTROL. The assimilation of AOD$_{550}$ (MASS), will increase the aerosol mixing ratio of all aerosols based on their extinction, but it will not account for their absorption. Thus AAOD will be increased along with AOD since more aerosols will be in the atmosphere.** Specifically, in the Amazon basin SSA$_{550}$ of the MASS experiment decreases by 0.032 in comparison to CONTROL, since the BC column burden becomes 4 times higher (FigureS 7b), while the difference of SSA$_{550}$ between POLDER and the model (spatiotemporal collocated points only) increases from -0.084 to -0.117 (FigureS 7c).

Figure 4 It's the wrong way around.
The caption in the figure has been corrected.

Figure 17f Why does MODIS-DB underestimate AODs where POLDER estimates AOD as about 0.1?
The majority of these AOD values are located in the desert area of Australia. The map below depicts the data points of Figure17f that satisfy the following AOD ranges 0.08 < POLDER < 0.12 AND 0.01 < MODIS-DB < 0.02. Different assumptions of the surface albedo by the retrieval algorithms may be causing this high differences between MODIS-DB (low) and POLDER (high) retrieved AOD. Although it is a very interesting topic, it is out of the scope of this paper, thus discussion was not added in the manuscript.